# USB: A Unified Semi-supervised Learning Benchmark for Classification

**Yidong Wang**[1,2,3*], **Hao Chen**[4*], **Yue Fan**[5*], **Wang Sun**[6], **Ran Tao**[4], **Wenxin Hou**[7],
**Renjie Wang**[8], **Linyi Yang**[2], **Zhi Zhou**[8], **Lan-Zhe Guo**[8], **Heli Qi**[9], **Zhen Wu**[8], **Yu-Feng Li**[8],
**Satoshi Nakamura**[9], **Wei Ye**[10], **Marios Savvides**[4], **Bhiksha Raj**[4], **Takahiro Shinozaki**[3],
**Bernt Schiele**[5], **Jindong Wang**[1†], **Xing Xie**[1], **Yue Zhang**[2†]

[1]Microsoft Research Asia, [2]Westlake University, [3]Tokyo Institute of Technology,
[4]Carnegie Mellon University, [5]Max-Planck-Institut für Informatik, [6]Tsinghua University,
[7]Microsoft STCA, [8]Nanjing University, [9]Nara Institute of Science and Technology, [10]Peking University

## Abstract

Semi-supervised learning (SSL) improves model generalization by leveraging massive unlabeled data to augment limited labeled samples. However, currently, popular SSL evaluation protocols are often constrained to computer vision (CV) tasks. In addition, previous work typically trains deep neural networks from scratch, which is time-consuming and environmentally unfriendly. To address the above issues, we construct a Unified SSL Benchmark (USB) for classification by selecting 15 diverse, challenging, and comprehensive tasks from CV, natural language processing (NLP), and audio processing (Audio), on which we systematically evaluate the dominant SSL methods, and also open-source a modular and extensible codebase for fair evaluation of these SSL methods. We further provide the pre-trained versions of the state-of-the-art neural models for CV tasks to make the cost affordable for further tuning. USB enables the evaluation of a single SSL algorithm on more tasks from multiple domains but with less cost. **Specifically, on a single NVIDIA V100, only 39 GPU days are required to evaluate FixMatch on 15 tasks in USB while 335 GPU days (279 GPU days on 4 CV datasets except for ImageNet) are needed on 5 CV tasks with TorchSSL.**

## 1   Introduction

Neural models give competitive results when trained using supervised learning on sufficient high-quality labeled data [1, 2, 3, 4, 5, 6, 7]. However, it can be laborious and expensive to obtain abundant annotations for model training [8, 9]. To address this issue, **semi-supervised learning (SSL)** emerges as an effective paradigm to improve model generalization with limited labeled data and massive unlabeled data [10, 11, 12, 13, 14, 15].

SSL has made remarkable progress in recent years [16, 17, 18, 19, 20, 21], yet there are still several limitations with the popular evaluation protocol in the literature [22, 20, 21]. First, existing benchmarks are mostly constrained to plain computer vision (CV) tasks (i.e., CIFAR-10/100, SVHN, STL-10, and ImageNet classification [22, 23, 20, 24, 21], as summarized in TorchSSL [21]), precluding consistent and diverse evaluation over tasks in natural language processing (NLP), audio processing (Audio), etc., where the lack of labeled data is a general issue and SSL has gained increasing research attention recently [25, 26, 27]. Second, the existing protocol (e.g., TorchSSL [21]) can be mainly time-consuming and environmentally unfriendly because it typically trains deep neural

---

*Equal contribution. Yidong Wang did this work during his internship at MSRA and Westlake University.
†Correspondence to: jindong.wang@microsoft.com, zhangyue@westlake.edu.cn.

36th Conference on Neural Information Processing Systems (NeurIPS 2022) Track on Datasets and Benchmarks.

Table 1: A summary of datasets and training cost used in (a) the existing popular protocol and (b) USB. USB largely reduces the training cost while providing a diverse, challenging, and comprehensive benchmark covering a wide range of datasets from various domains. Training cost is estimated by using FixMatch [20] on a single NVIDIA V100 GPU from Microsoft Azure Machine Learning platform, except for ImageNet where 4 V100s are used. Experiments in (a) follow the settings in [21]. More results with different pre-trained backbones are available in Appendix D.

(a) TorchSSL [21]

| Domain & Backbone | Dataset | Classification Task | Hours × Settings × Seeds | Total GPU Hours | Total GPU Hours w/o ImageNet |
|---|---|---|---|---|---|
| CV, ResNets | CIFAR-10 | Natural Image | 110 × 3 × 3 | 8031 GPU Hours (335 GPU Days) | 6687 GPU Hours (279 GPU Days) |
| | CIFAR-100 | Natural Image | 300 × 3 × 3 | | |
| | SVNH | Digital | 108 × 3 × 3 | | |
| | STL-10 | Natural Image | 225 × 3 × 3 | | |
| | ImageNet | Natural Image | 336 hours × 4 GPUs | | |

(b) USB

| Domain & Backbone | Dataset | Classification Task | Hours × Settings × Seeds | Total GPU Hours |
|---|---|---|---|---|
| CV, ViTs | CIFAR-100 | Natural Image | 11 × 2 × 3 | 924 GPU Hours (39 GPU Days) |
| | STL-10 | Natural Image | 18 × 2 × 3 | |
| | EuroSAT | Satellite Image | 10 × 2 × 3 | |
| | TissueMNIST | Medical Image | 8 × 2 × 3 | |
| | Semi-Aves | Fine-grained, Long-tailed Natural Image | 13 × 1 × 3 | |
| NLP, Bert | IMDB | Movie Review Sentiment | 8 × 2 × 3 | |
| | AG News | News Topic | 6 × 2 × 3 | |
| | Amazon Review | Product Review Sentiment | 8 × 2 × 3 | |
| | Yahoo! Answer | QA Topic | 7 × 2 × 3 | |
| | Yelp Review | Restaurant Review Sentiment | 8 × 2 × 3 | |
| Audio, Wave2Vec 2.0 and HuBert | GTZAN | Music Genre | 12 × 2 × 3 | |
| | UrtraSound8k | Urban Sound Event | 15 × 2 × 3 | |
| | FSDnoisy18k | Sound Event | 17 × 1 × 3 | |
| | Keyword Spotting | Keyword | 10 × 2 × 3 | |
| | ESC-50 | Environmental Sound Event | 18 × 2 × 3 | |

models from scratch [28, 23, 29, 20, 24, 21]. Specifically, as shown in Table 1a, it takes about 335 GPU days (279 GPU days without ImageNet) to evaluate FixMatch [20] with TorchSSL [21]. Such a high cost can make it unaffordable for research labs (particularly in academia) to conduct SSL research. Recently, the pre-training and fine-tuning paradigm [30, 31, 32, 33] achieves promising results. Compared with training from scratch, pre-training has much reduced cost in SSL. However, there are relatively few benchmarks that offer a fair test bed for SSL with the pre-trained versions of neural models.

To address the above issues and facilitate general SSL research, we propose **USB: a Unified SSL Benchmark** for classification [3]. USB offers a *diverse* and *challenging* benchmark across five CV datasets, five NLP datasets, and five Audio datasets (Table 1b), enabling consistent evaluation over multiple tasks from different domains. Moreover, USB provides comprehensive evaluations of SSL algorithms with even fewer labeled data compared with TorchSSL, as the performance gap between SSL algorithms diminishes when the amount of labeled samples becomes large. Benefiting from the rapidly developed neural architectures, we introduce pre-trained Transformers [4] into SSL instead of training ResNets [1] from scratch to reduce the training cost for CV tasks. Specifically, we find that using pre-trained Vision Transformers (ViT) [34] can largely reduce the number of training iterations (e.g., by 80% from 1,000k to 200k on CV tasks) without hurting the performance, and most SSL algorithms achieve even better performance with less training iterations.

As illustrated in Table 1b, using USB, we spend only **39 GPU days** to evaluate the performance of an SSL algorithm (i.e., FixMatch) on a single NVIDIA V100 over these **15 datasets**, in contrast to TorchSSL, which costs about **335 GPU days** on only **5 CV datasets** (279 GPU days on 4 CV datasets except for ImageNet). To further facilitate SSL research, we open-source the codebase and pre-trained models [4] for unified and consistent evaluation of SSL methods. In addition, we also provide config files that contain all the hyper-parameters to easily reproduce our results reported in

---

[3]The word 'unified' means the unification of different algorithms on various application domains.

[4]https://github.com/microsoft/Semi-supervised-learning. We also provide the training logs of the experiments in this paper. Note that the results and training logs will be continuously updated/provided if we reorganize the codes for better use or add more algorithms and datasets. Microsoft Research Asia (MSRA) will provide both the support and resources for future updates.

Table 2: The comparison between USB and other related benchmarks.

| Benchmark | # SSL algorithms | Domian | # Tasks | Pre-trained | Training hours using FixMatch |
|---|---|---|---|---|---|
| Realistic SSL evaluation [22] | 4 | CV | 3 | ✗ | - |
| TorchSSL [21] | 9 | CV | 5 | ✗ | 6687 |
| USB | 14 | CV, NLP, Audio | 15 | ✓ | 924 |

this work. We obtain some interesting findings by evaluating 14 SSL algorithms (Section 5.4): (1) introducing diverse tasks from diverse domains can be beneficial to comprehensive evaluation of an SSL algorithm; (2) pre-training is more efficient and can improve the generalization; (3) unlabeled data do not consistently improve the performance especially when labeled data is scarce.

To conclude, our contributions are three-fold:

- We propose USB: a unified and challenging semi-supervised learning benchmark for classification with 15 tasks on CV, NLP, and Audio for fair and consistent evaluations. To our humble knowledge, we are the first to discuss whether current SSL methods that work well on CV tasks generalize to NLP and Audio tasks.
- We provide an environmentally friendly and low-cost evaluation protocol with pre-training & fine-tuning paradigm, reducing the cost of SSL experiments. The advantages of USB as compared to other related benchmarks are shown in Table 2.
- We implement 14 SSL algorithms and open-source a modular codebase and config files for easy reproduction of the reported results in this work. we also provide documents and tutorials for easy modification. Our codebase is extensible and open for continued development through community effort, where we expect new algorithms, models, config files and results are constantly added.

## 2 Related Work

Deep semi-supervised learning originates from $\Pi$ model [35], where it solves the task of image classification by using consistency regularization that forces the model to output similar predictions when fed two augmented versions of the same unlabeled data. Subsequent methods can be classified as the variants of $\Pi$ model, where the difference lies in enforcing the consistency between model perturbation [36], data perturbation [37, 29], and exploiting unlabeled data [20, 21]. Since the best results in both CV and NLP are given by such algorithms, we choose them as typical representative methods in USB. While most SSL methods have seen their use in CV tasks, NLP has witnessed recent growth in SSL solutions [29, 25]. However, only some of the popular methods [29] in CV have been used in the NLP literature, probably because other methods give lower results or have not been investigated. This gives us motivation for evaluation of SSL methods on various domains in USB.

As shown in Table 2, related benchmarks include Realistic SSL evaluation [22] and TorchSSL [21]. Realistic SSL evaluation [22] has 4 SSL algorithms and 3 CV classification tasks and TorchSSL has 9 SSL algorithms and 5 CV classification tasks. Both of them are no longer maintained/updated. Thus it is of significance to build an SSL community that can continuously update SSL algorithms and neural models to boost the development of SSL. Besides, previous benchmarks mainly train the models from scratch, which is computation expensive and time consuming, since SSL algorithms are known to be difficult to converge [38]. In USB, we consider using pre-trained models to boost the performance while being more efficient and friendly to researchers.

In the following, we will first introduce the tasks, datasets, algorithms, and benchmark results of USB. Then, the codebase structure of USB will be presented in Section 6.

## 3 Tasks and Datasets

USB consists of 15 datasets from CV, NLP, and Audio domains. Every dataset in USB is under a permissive license that allows usage for research purposes. The datasets are chosen based on the following considerations: (1) the tasks should be diverse and cover multiple domains; (2) the tasks should be challenging, leaving room for improvement; (3) the training is reasonably environmentally friendly and affordable to research labs (in both the industry and academia).

Table 3: Details of the datasets in USB. Two *#Label per class* settings are chosen for each dataset except Semi-Aves and FSDnoisy18k, which have long-tailed distributed data. Labeled data are sampled from the training data for each dataset except STL-10, Semi-Aves, and FSDNoisy18k, where the split of labeled and unlabeled data is pre-defined (e.g. 5,959 labeled images and 26,640 unlabeled images in Semi-Aves). Following [20, 21], validation data are not provided for CV datasets. The NLP validation data are sampled from the original training datasets. All test sets are kept unchanged.

| Domain | Dataset | #Label per class | #Training data | #Validation data | #Test data | #Class |
|---|---|---|---|---|---|---|
| CV | CIFAR-100 | 2 / 4 | 50,000 | - | 10,000 | 100 |
| | STL-10 | 4 / 10 | 5,000 / 100,000 | - | 8,000 | 10 |
| | EuroSat | 2 / 4 | 16,200 | - | 5,400 | 10 |
| | TissueMNIST | 10 / 50 | 165,466 | - | 47,280 | 8 |
| | Semi-Aves | 15-53 | 5,959 / 26,640 | - | 4,000 | 200 |
| NLP | IMDB | 10 / 50 | 23,000 | 2,000 | 25,000 | 2 |
| | Amazon Review | 50 / 200 | 250,000 | 25,000 | 65,000 | 5 |
| | Yelp Review | 50 / 200 | 250,000 | 25,000 | 50,000 | 5 |
| | AG News | 10 / 50 | 100,000 | 10,000 | 7,600 | 4 |
| | Yahoo! Answer | 50 / 200 | 500,000 | 50,000 | 60,000 | 10 |
| Audio | Keyword Spotting | 5 / 20 | 18,538 | 2,577 | 2,567 | 10 |
| | ESC-50 | 5 / 10 | 1,200 | 400 | 400 | 50 |
| | UrbanSound8k | 10 / 40 | 7,079 | 816 | 837 | 10 |
| | FSDnoisy18k | 52-171 | 1,772 / 15,813 | - | 947 | 20 |
| | GTZAN | 10 / 40 | 7,000 | 1,500 | 1,500 | 10 |

## 3.1 CV Tasks

The details of the CV datasets are shown in Table 3. We include CIFAR-100 [39] and STL-10 [40] from TorchSSL since they are still challenging. The TissueMNIST [41, 42], EuroSAT [43, 44], and Semi-Aves [45] are datasets in the domains of medical images, satellite images, and fine-grained natural images. CIFAR-10 [39] and SVHN [46] in TorchSSL are not included in USB because the state-of-the-art SSL algorithms [29, 20, 24] have achieved similar performance on these datasets to fully-supervised training with abundant fully labeled training data [5]. SSL algorithms have a relatively large room for improvement on all chosen CV datasets in USB. More details of these CV datasets in USB can be found in Appendix E.1.

## 3.2 NLP Tasks

The detailed dataset statistics of NLP tasks in USB are described in Table 3. We mostly followed previous work in the NLP literature, and thus the existing datasets in USB cover most test sets used in the existing work [25, 48, 29]. We include widely used IMDB [49], AG News [50], and Yahoo! Answer [51] from the previous protocol [25, 48, 29], which are still challenging for SSL. Since IMDB is a binary sentiment classification task, we further add Amazon Review [52] and Yelp Review [53] to evaluate SSL algorithms on more fine-grained sentiment classification tasks. DBpedia is removed from the previous protocol [25, 48, 29] because we find that the state-of-the-art SSL algorithms have achieved similar performance on it when compared with fully-supervised training. For all tasks in NLP, we obtain the labeled datasets, unlabeled datasets, and validation sets by randomly sampling from their original training datasets while keeping the original test datasets unchanged, mainly following previous work [25, 48]. More details are in Appendix E.2.

## 3.3 Audio Tasks

USB includes five audio classification datasets as shown in Table 3. We choose the tasks to cover different domains such as urban sound (UrbanSound8k [54], ESC-50 [55], and FSDNoisy18k [56]), human sound (Keyword Spotting [57]), and music (GTZAN) [58]. All chosen datasets are challenging even for state-of-the-art SSL algorithms. For example, FSDNoisy18k is a realistic dataset containing a small labeled set and a large unlabeled set. To the best of our knowledge, we are the first to systematically evaluate SSL algorithms on Audio tasks. Although there is a concurrent work [27], our study includes more algorithms and more datasets than [27]. More details are in Appendix E.3.

---

[5]We highly recommend reporting ImageNet [8] results since it is a reasonable dataset for hill-climbing [20, 47, 21]. We also report and discuss ImageNet results in Appendix C.

Table 4: Essential components used in 14 SSL algorithms supported in USB. PL, CR, Dist. Align., and W-S Aug., MSE, CE are the abbreviations for Pseudo Labeling, Consistency Regularization, Distribution Alignment, Weak-Strong Augmentation, Mean Squared Error, and Cross-Entropy, respectively. PL denotes hard 'one-hot' labels adopted in CR Loss.

| Algorithm | PL | CR Loss | Thresholding | Dist. Align. | Self-supervised | Mixup | W-S Aug. |
|---|---|---|---|---|---|---|---|
| Π-Model | | MSE | | | | | |
| Pseudo Labeling | ✓ | CE | | | | | |
| Mean Teacher | | MSE | | | | | |
| VAT | | CE | | | | | |
| MixMatch | | MSE | | | | ✓ | |
| ReMixMatch | | CE | | ✓ | Rotation | ✓ | ✓ |
| UDA | | CE | ✓ | | | | ✓ |
| FixMatch | ✓ | CE | ✓ | | | | ✓ |
| Dash | ✓ | CE | ✓ | | | | ✓ |
| CoMatch | ✓ | CE | ✓ | ✓ | Contrastive | | ✓ |
| CRMatch | ✓ | CE | ✓ | | Rotation | | ✓ |
| FlexMatch | ✓ | CE | ✓ | | | | ✓ |
| AdaMatch | ✓ | CE | ✓ | ✓ | | | ✓ |
| SimMatch | ✓ | CE | ✓ | ✓ | Contrastive | | ✓ |

## 4 SSL Algorithms

We implement 14 SSL algorithms in the codebase for USB, including Π model [35], Pseudo Labeling [59], Mean Teacher [36], VAT [37], MixMatch [28], ReMixMatch [23], UDA [29], FixMatch [20], Dash [24], CoMatch [60], CRMatch [61], FlexMatch [21], AdaMatch [62], and SimMatch [47], all of which exploit unlabeled data by encouraging invariant predictions to input perturbations [13, 14, 63, 64, 65, 66, 67]. Such consistency regularization methods give the strongest performance in SSL since the model is robust to different perturbed versions of unlabeled data, satisfying the smoothness and low-density assumptions in SSL [68].

The above SSL algorithms use Cross-Entropy (CE) loss on labeled data but differ in the way on unlabeled data. As shown in Table 4, Pseudo Labeling [59] turns the predictions of the unlabeled data into hard 'one-hot' labels and treats the 'one-hot' pseudo-labels as the supervision signals. Thresholding reduces the noisy pseudo labels by masking out the unlabeled samples whose maximum probabilities are smaller than the pre-defined threshold. Distribution Alignment aims to correct the output distribution to make it more in line with the target distribution (e.g., uniform distribution). Self-supervised learning, Mixup, and Stronger augmentations techniques also can help learn better representation. More details of these algorithms can be found in Appendix F. We summarize the key components exploited in the implemented consistency regularization based algorithms in Table 4.

## 5 Benchmark Results

[6] For CV tasks, we follow [21] to report the best number of all checkpoints to avoid unfair comparisons caused by different convergence speeds. For NLP and Audio tasks, we choose the best model using the validation datasets and then evaluate it on the test datasets. In addition to mean error rate over the tasks, we use Friedman rank [69, 70] to fairly compare the performance of different algorithms in various settings:

$$\text{rank}_F = \frac{1}{m} \sum_{i=1}^{m} \text{rank}_i,$$

where $m$ is the number of evaluation settings (i.e., how many experimental settings we use, e.g., $m = 9$ in Table 5), and $\text{rank}_i$ is the rank of an SSL algorithm in the $i$-th setting. We re-rank all algorithms to give final ranks based on their Friedman rankings. Note that all ranks are in ascending order because the lower error rate corresponds to a better performance. The experimental setup is detailed in Appendix G. Note that 'supervised' denotes training with the partially chosen labeled data while 'fully-supervised' refers to training using all data with full annotations in our reported results.

---

[6]**Note that all experimental results and training logs will be continuously updated in** `https://github.com/microsoft/Semi-supervised-learning`. **Please refer to the latest results for comparison.**

Table 5: Error rate (%) and Rank with CV tasks in USB. For Semi-Aves and STL10, as they have unlabeled sets, we do not report the fully-supervised results. We follow [20, 21, 29] to show error rates as default.

| Dataset | CIFAR-100 | | STL-10 | | Euro-SAT | | TissueMNIST | | Semi-Aves | Friedman | Final | Mean |
|---|---|---|---|---|---|---|---|---|---|---|---|---|
| # Label | 200 | 400 | 20 | 40 | 20 | 40 | 80 | 400 | 5,959 | rank | rank | error rate |
| Fully-Supervised | 8.44±0.07 | 8.44±0.07 | - | - | 0.94±0.07 | 0.89±0.05 | 29.15±0.13 | 29.10±0.02 | - | - | - | - |
| Supervised | 35.63±0.36 | 26.08±0.50 | 47.02±1.48 | 26.02±0.72 | 27.12±1.26 | 16.90±1.48 | 59.91±2.93 | 54.10±1.52 | 41.55±0.29 | - | - | - |
| Π-model | 36.24±0.27 | 26.49±0.64 | 44.38±1.59 | 25.76±2.37 | 24.51±1.02 | 11.58±1.32 | 56.79±5.91 | **47.50±1.71** | 39.23±0.36 | 10.11 | 11 | 34.72 |
| Pseudo-Labeling | 33.16±1.20 | 25.29±0.67 | 45.13±4.08 | 26.20±1.53 | 23.64±0.90 | 15.61±2.51 | 56.22±4.01 | 50.36±1.62 | 40.13±0.09 | 9.89 | 10 | 35.08 |
| Mean Teacher | 35.61±0.38 | 25.97±0.37 | 39.94±1.99 | 20.16±1.25 | 26.51±1.15 | 17.05±2.07 | 61.40±2.48 | 55.22±2.06 | 38.52±0.27 | 10.89 | 14 | 35.60 |
| VAT | 31.61±1.37 | 21.29±0.32 | 52.03±0.48 | 23.10±0.72 | 24.77±1.94 | 9.30±1.23 | 58.50±6.41 | 51.31±1.66 | 39.00±0.30 | 10.11 | 12 | 34.55 |
| MixMatch | 37.43±0.58 | 26.17±0.24 | 48.98±1.41 | 25.56±3.00 | 29.86±2.89 | 16.39±3.17 | 55.73±2.29 | 49.08±1.06 | 37.22±0.15 | 10.11 | 12 | 36.27 |
| ReMixMatch | **20.85±1.42** | **16.80±0.59** | 30.61±3.47 | 18.33±1.98 | **4.53±1.60** | **4.10±0.37** | 59.29±5.16 | 52.92±3.93 | **30.40±0.33** | 4.00 | 1 | 26.43 |
| UDA | 30.75±1.03 | 19.94±0.32 | 39.22±2.87 | 23.59±2.97 | 11.15±1.20 | 5.99±0.75 | 55.88±3.26 | 51.42±2.05 | 31.74±0.33 | 6.89 | 7 | 30.05 |
| FixMatch | 30.45±0.65 | 19.48±0.93 | 42.06±3.94 | 24.05±1.79 | 12.48±2.57 | 6.41±1.64 | 55.95±4.06 | 50.93±1.23 | 31.74±0.33 | 6.56 | 6 | 30.39 |
| Dash | 30.19±1.34 | 18.90±0.42 | 43.34±1.46 | 25.90±0.35 | 9.44±0.75 | 7.00±1.39 | 57.00±2.81 | 50.93±1.54 | 32.56±0.39 | 7.44 | 9 | 30.58 |
| CoMatch | 35.68±0.54 | 26.10±0.09 | **29.70±1.17** | 21.46±1.34 | 5.25±0.49 | 4.89±0.86 | 57.15±3.46 | 51.83±0.71 | 41.39±0.16 | 7.22 | 8 | 30.38 |
| CRMatch | 29.43±1.11 | 18.50±0.26 | 30.55±2.01 | **17.43±1.96** | 14.52±1.34 | 7.00±0.69 | **54.84±3.05** | 51.10±1.59 | 31.97±0.10 | 4.67 | 2 | 28.37 |
| FlexMatch | 27.08±0.90 | 17.67±0.66 | 37.58±2.97 | 23.40±1.50 | 7.07±2.32 | 5.58±0.57 | 57.23±2.50 | 52.06±1.78 | 33.09±0.16 | 6.44 | 5 | 28.97 |
| AdaMatch | 21.27±1.04 | 17.01±0.55 | 36.25±1.89 | 23.30±0.73 | 5.70±0.37 | 4.92±0.87 | 57.87±4.47 | 52.28±0.79 | 31.54±0.10 | 5.22 | 3 | 27.79 |
| SimMatch | 23.26±1.25 | 16.82±0.40 | 34.12±1.63 | 22.97±2.04 | 6.88±1.77 | 5.86±1.07 | 57.91±4.60 | 51.14±1.83 | 34.14±0.30 | 5.44 | 4 | 28.12 |

Table 6: Error rate (%) and Rank with NLP tasks in USB.

| Dataset | IMDB | | AG News | | Amazon Review | | Yahoo! Answer | | Yelp Review | | Friedman | Final | Mean |
|---|---|---|---|---|---|---|---|---|---|---|---|---|---|
| # Label | 20 | 100 | 40 | 200 | 250 | 1000 | 500 | 2000 | 250 | 1000 | rank | rank | error rate |
| Fully-Supervised | 5.87±0.01 | 5.84±0.12 | 5.74±0.30 | 5.64±0.05 | 36.81±0.05 | 36.88±0.19 | 26.25±1.07 | 25.55±0.43 | 31.74±0.20 | 32.70±0.58 | - | - | - |
| Supervised | 20.63±3.13 | 13.47±0.55 | 15.01±1.21 | 13.00±1.00 | 51.74±0.63 | 47.34±0.66 | 37.10±1.22 | 33.56±0.08 | 50.27±0.51 | 46.96±0.42 | - | - | - |
| Π-Model | 49.02±1.37 | 27.57±15.85 | 46.84±6.20 | 13.44±0.76 | 73.53±6.92 | 48.27±0.48 | 41.37±2.15 | 32.96±0.16 | 73.35±2.31 | 52.02±1.48 | 11.80 | 12 | 45.84 |
| Pseudo-Labeling | 26.38±4.04 | 21.38±1.34 | 23.86±7.63 | 12.29±0.40 | 53.00±1.48 | 46.49±0.45 | 38.60±1.09 | 33.44±0.20 | 55.70±0.95 | 47.72±0.37 | 10.60 | 11 | 35.89 |
| Mean Teacher | 21.27±3.72 | 14.11±1.77 | 14.98±1.10 | 13.23±1.12 | 51.67±0.45 | 47.51±0.24 | 36.97±1.02 | 33.43±0.22 | 51.07±1.44 | 46.61±0.34 | 9.30 | 10 | 33.09 |
| VAT | 32.59±4.69 | 14.42±2.53 | 15.00±1.12 | 11.59±0.94 | 50.38±0.83 | 46.04±0.28 | 35.16±0.74 | 31.53±0.41 | 52.76±0.87 | 45.53±0.13 | 8.40 | 8 | 33.50 |
| UDA | 9.36±1.26 | 8.33±0.61 | 18.73±2.68 | 12.34±1.90 | 52.48±1.20 | 45.51±0.61 | 35.31±0.43 | 32.01±0.68 | 58.22±0.40 | 42.18±0.68 | 8.70 | 9 | 31.45 |
| FixMatch | 8.20±0.29 | **7.36±0.07** | 22.80±5.18 | 11.43±0.65 | 47.85±1.22 | 43.73±0.45 | 34.15±0.94 | 30.76±0.53 | 50.34±0.40 | 41.99±0.58 | 5.60 | 7 | 29.86 |
| Dash | 8.93±1.27 | 7.97±0.53 | 19.30±6.73 | 11.20±1.12 | 47.79±1.03 | 43.52±0.07 | 35.10±1.36 | 30.51±0.47 | 47.99±1.05 | 41.59±0.61 | 5.10 | 6 | 29.39 |
| CoMatch | 7.36±0.26 | 7.41±0.20 | 13.25±1.31 | 11.61±0.42 | 48.98±1.20 | 44.37±0.25 | 33.48±0.67 | **30.19±0.22** | 46.49±1.42 | 41.11±0.53 | 3.80 | 3 | 28.43 |
| CRMatch | 7.88±0.24 | 7.68±0.35 | 13.35±1.06 | 11.36±1.04 | 46.23±0.85 | 43.69±0.48 | 33.07±0.68 | 30.62±0.47 | 46.61±1.02 | 41.80±0.77 | 3.70 | 2 | 28.23 |
| FlexMatch | 7.35±0.10 | 7.80±0.24 | 16.90±6.76 | 11.43±0.91 | **45.75±1.21** | 43.14±0.82 | 35.81±1.09 | 31.42±0.41 | 46.37±0.74 | **40.86±0.74** | 4.10 | 5 | 28.68 |
| AdaMatch | 9.62±1.26 | 7.81±0.46 | **12.92±1.53** | **11.03±0.62** | 46.75±1.23 | 43.50±0.67 | **32.97±0.43** | 30.82±0.29 | 48.16±0.80 | 41.71±1.08 | 4.00 | 4 | 28.53 |
| SimMatch | **7.24±0.02** | 7.44±0.20 | 14.80±0.57 | 11.12±0.15 | 47.27±1.73 | **43.09±0.50** | 34.15±0.91 | 30.64±0.42 | **46.40±1.71** | 41.24±0.17 | 2.90 | 1 | 28.34 |

The results for the 14 SSL algorithms on the datasets from CV, NLP, and Audio are shown in Table 5, Table 6, and Table 7, respectively. We adopt the pre-trained Vision Transformers (ViT) [4, 34, 30, 71] instead of training ResNets [1] from scratch for CV tasks. For NLP, we adopt Bert [30]. Wav2Vec 2.0 [71] and HuBert [32] are used for Audio.

## 5.1 CV Results

The results are illustrated in Table 5. Thanks to the good initialization of representation on unlabeled data given by the pre-trained ViT, SSL algorithms, even without using thresholding techniques, often achieve much better performance than the previous performance shown in TorchSSL [21]. Among all the SSL algorithms, ReMixMatch [23] ranks at the first and outperforms other SSL algorithms, due to the usage of Mixup, Distribution Alignment, and rotation self-supervised loss. Its superiority is especially demonstrated in the evaluation of Semi-Aves, a long-tailed and fine-grained CV dataset that is more realistic. Notice that SSL algorithms with self-supervised feature loss generally perform well than other SSL algorithms, e.g., CRMatch [61] and SimMatch [47] rank second and fourth respectively. Adaptive thresholding algorithms also demonstrate their effectiveness, e.g., AdaMatch [62] and FlexMatch [21] rank at third and fifth respectively. While better results of the evaluated SSL algorithms are obtained on CIFAR-100, Euro-SAT, and Semi-Aves, we also observe that the performance is relatively lower on STL-10 and TissueMNIST. The reason for lower performance on STL-10 might result from the usage of the self-supervised pre-trained model [33], rather than the supervised pre-trained model is used in other settings. Since TissueMNIST is a medial-related dataset, the biased pseudo-labels might produce a destructive effect that impedes training and leads to bad performance. The de-biasing of pseudo-labels and safe semi-supervised learning would be interesting topics in future work, especially for medical applications of SSL algorithms.

## 5.2 NLP Results

The results of NLP tasks are demonstrated in Table 6. The overall ranking of SSL algorithms in NLP is similar to that in CV. However, the SSL algorithm that works well in NLP does not always

Table 7: Error rate (%) and Rank with Audio tasks in USB. Fully-supervised result is not reported for FSDNoisy18k due to the unknown labels of its unlabeled set.

| Dataset | GTZAN | | UrbanSound8k | | Keyword Spotting | | ESC-50 | | FSDnoisy | Friedman | Final | Mean |
|---|---|---|---|---|---|---|---|---|---|---|---|---|
| # Label | 100 | 400 | 100 | 400 | 50 | 100 | 250 | 500 | 1,772 | rank | rank | error rate |
| Fully-Supervised | 5.98±0.32 | 5.98±0.32 | 16.65±1.71 | 16.61±1.71 | 2.12±0.11 | 2.25±0.02 | 26.00±2.13 | 26.00±2.13 | - | - | - | - |
| Supervised | 52.16±1.83 | 31.53±0.52 | 40.42±1.00 | 28.55±1.90 | 6.80±1.16 | 5.25±0.56 | 51.58±1.12 | 35.67±0.42 | 35.20±1.50 | - | - | - |
| Π-Model | 74.07±0.62 | 33.18±3.64 | 54.24±6.01 | 25.89±1.51 | 64.39±4.10 | 25.48±4.94 | 47.25±1.14 | 36.00±1.62 | 35.73±0.87 | 10.67 | 12 | 44.03 |
| Pseudo-Labeling | 57.29±2.80 | 33.93±0.69 | 42.09±2.41 | 27.00±1.34 | 7.82±1.64 | 5.16±0.14 | 49.33±2.52 | 35.58±1.05 | 35.34±1.60 | 10.00 | 10 | 32.62 |
| Mean Teacher | 51.40±3.48 | 31.60±1.46 | 41.70±3.39 | 28.91±0.93 | 5.95±0.44 | 5.39±0.42 | 50.25±1.95 | 37.33±1.20 | 35.83±1.22 | 10.33 | 11 | 32.04 |
| VAT | 79.51±1.99 | 35.38±7.80 | 49.62±2.42 | 27.68±1.39 | **2.18±0.08** | **2.23±0.08** | 46.42±1.90 | 36.92±2.25 | 32.07±1.05 | 8.33 | 9 | 34.67 |
| UDA | 46.56±8.69 | 23.62±0.63 | 37.28±3.17 | 20.27±1.58 | 2.52±0.15 | 2.62±0.10 | 42.75±0.89 | 33.50±1.95 | 30.80±0.47 | 6.33 | 7 | 26.66 |
| FixMatch | 36.04±4.57 | 22.09±0.65 | 36.12±4.26 | 21.43±2.88 | 4.84±3.57 | 2.38±0.03 | **37.75±3.19** | 30.67±1.05 | 30.31±1.08 | 4.00 | 3 | 24.63 |
| Dash | 47.00±3.65 | 23.42±0.83 | 42.02±5.02 | 22.26±0.89 | 5.70±4.40 | 2.52±0.16 | 48.17±1.16 | 32.75±2.27 | 33.19±0.95 | 7.56 | 8 | 28.56 |
| CoMatch | 36.93±1.23 | 22.20±1.39 | **30.59±2.45** | 21.35±1.49 | 11.39±0.85 | 9.44±1.52 | 40.17±2.08 | **29.83±1.31** | 27.63±1.35 | 5.11 | 6 | 25.50 |
| CRMatch | 40.58±3.97 | 22.64±1.22 | 39.47±4.66 | 20.11±2.63 | 2.40±0.13 | 2.49±0.08 | 42.67±0.51 | 33.58±1.93 | 30.45±1.52 | 5.00 | 5 | 26.04 |
| FlexMatch | 34.60±4.07 | 21.82±1.17 | 40.18±2.73 | 22.82±3.10 | 2.42±0.08 | 2.57±0.25 | 39.58±0.59 | 29.92±1.85 | **26.36±0.55** | 4.11 | 4 | 24.47 |
| AdaMatch | **31.38±0.41** | **20.73±0.67** | 35.76±6.39 | 21.15±1.22 | 2.49±0.08 | 2.49±0.10 | 39.17±1.74 | 31.33±1.23 | 27.95±0.74 | 2.89 | 1 | 23.61 |
| SimMatch | 32.42±2.18 | 20.80±0.77 | 31.70±6.05 | **19.55±1.89** | 2.57±0.08 | 2.53±0.22 | 39.92±2.35 | 32.83±1.43 | 28.16±0.87 | 3.67 | 2 | 23.39 |

guarantee good performance in CV, which shows that the performance of SSL algorithms will be affected largely by data domains. For example, SimMatch which ranks first in NLP does not have the best performance in CV tasks (ranks fourth). The ranking of CoMatch is also increased in NLP, compared to that in CV. A possible reason is the different pre-training in backbones. For BERT, a masked language modeling objective is used during pre-training [30], thus the self-supervised feature loss might further improve the representation during fine-tuning with SSL algorithms. We observe that adaptive thresholding methods, such as FlexMatch and AdaMatch, consistently achieve good performance on both CV and NLP, even without self-supervised loss. Note that we do not evaluate MixMatch and ReMixMatch on NLP and Audio tasks because we find that mixing sentences with different lengths harms the model's performance.

## 5.3 Audio Results

The results of Audio tasks are shown in Table 7. AdaMatch outperforms other algorithms in Audio tasks, while SimMatch demonstrates a similar performance to AdaMatch. An interesting finding is that CRMatch performs well on CV and NLP tasks, but badly in Audio tasks. We hypothesize that this is partially due to the noisy nature of the raw data in audio tasks. Except for Keyword Spotting, the gap between the performance of fully-supervised learning and that of SSL algorithms in Audio tasks is larger than in CV and NLP tasks. The reason behind this is probably that we exploit models that take waveform as input, rather than Mel spectrogram. Raw waveform might contain more noisy information that would be harmful to semi-supervised training. We identify exploring audio models based on Mel spectrogram as one of the future directions of USB.

## 5.4 Discussion

The evaluation results of SSL algorithms using USB are generally consistent with the results reported by previous work [22, 28, 23, 29, 20, 21]. However, using USB, we still provide some distinct quantitative and qualitative analysis to inspire the community. This section aims to answer the following questions: (1) Why should we evaluate an SSL algorithm on diverse tasks across domains? (2) Which option is better in the SSL scenario, training from scratch or using pre-training? (3) Does SSL consistently guarantee the performance improvement when using the state-of-the-art neural models as the backbones?

**Performance Comparisons** Table 8 shows the performance comparison of SSL algorithms in CV, NLP and Audio tasks. Although the ranking of each SSL algorithm in each domain is roughly close, the differences between ranks of SSL algorithms in different domains cannot be ignored. For example, FixMatch, CoMatch and CrMatch show large difference ($Rank_{max} - Rank_{min} \geq 4$) on the ranks across domains, which indicates that NLP and Audio tasks may have different characteristics compared with CV tasks that are more amenable to certain types of SSL algorithms compared with others. From the task perspective, it is important to consider such characteristics for guiding the choice of SSL methods. From the benchmarking perspective, it is useful to introduce diverse tasks from multiple domains when evaluating an SSL algorithm.

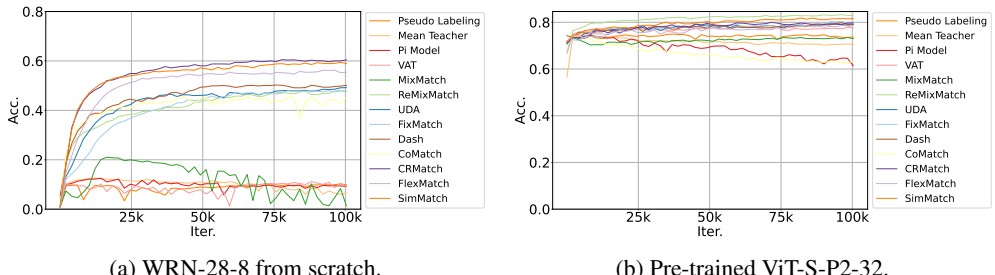

(a) WRN-28-8 from scratch.
(b) Pre-trained ViT-S-P2-32.

Figure 1: Comparison of test accuracy of SSL algorithms on CIFAR-100 with 400 labels. (a) Existing protocol which trains WRN-28-8 from scratch; (b) USB CV protocol which trains ImageNet-1K pre-trained ViT-S-P2-32, where S denotes small, P denotes patch size, and 32 is input image size.

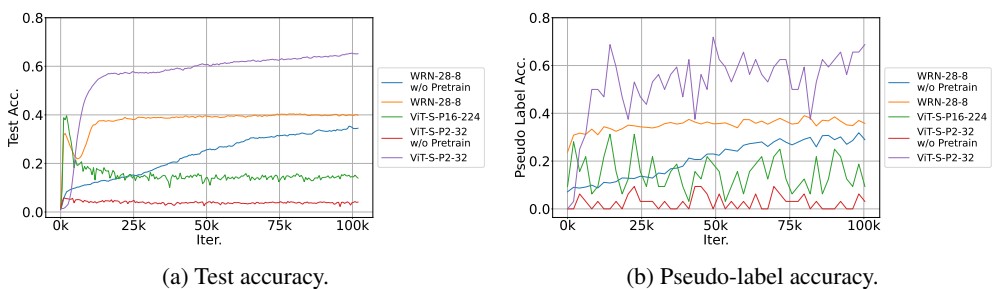

(a) Test accuracy.
(b) Pseudo-label accuracy.

Figure 2: Pre-training ablation on CIFAR-400 with 400 labels. Test and pseudo-label accuracy are compared with WRN-28-8 without pre-training, pre-trained WRN-28-8, pre-trained ViT-S-P16-224, ViT-S-P2-32 without pre-training, and pre-trained ViT-S-P2-32.

Table 8: Final ranks of SSL algorithms. Note that the rank for CV tasks here is different from the ones in Table 5 because we ignore MixMatch and ReMixMatch here to remove the effects of their missing ranks in NLP and Audio.

| | Π-Model | Pseudo-Labeling | Mean Teacher | VAT | UDA | FixMatch | Dash | CoMatch | CRMatch | FlexMatch | AdaMatch | SimMatch |
|---|---|---|---|---|---|---|---|---|---|---|---|---|
| CV | 10 | 9 | 12 | 11 | 6 | 5 | 8 | 7 | 1 | 4 | 2 | 3 |
| NLP | 12 | 11 | 10 | 8 | 9 | 7 | 6 | 3 | 2 | 5 | 4 | 1 |
| Audio | 12 | 10 | 11 | 9 | 7 | 3 | 8 | 6 | 5 | 4 | 1 | 2 |
| $\text{Rank}_{max} - \text{Rank}_{min}$ | 2 | 2 | 2 | 3 | 3 | **4** | 2 | **4** | **4** | 1 | 3 | 2 |

Table 9: This table shows how many times an SSL algorithm is worse than supervised training, where the numbers of total settings are 9, 10, and 9 for CV, NLP, and Audio respectively.

| | Π-Model | Pseudo-Labeling | Mean Teacher | VAT | MixMatch | ReMixMatch | UDA | FixMatch | Dash | CoMatch | CRMatch | FlexMatch | AdaMatch | SimMatch |
|---|---|---|---|---|---|---|---|---|---|---|---|---|---|---|
| CV | 2 | 1 | 3 | 1 | 4 | 0 | 0 | 0 | 0 | 2 | 0 | 0 | 0 | 0 |
| NLP | 9 | 7 | 5 | 3 | - | - | 2 | 1 | 1 | 0 | 0 | 1 | 0 | 0 |
| Audio | 7 | 5 | 6 | 4 | - | - | 0 | 0 | 1 | 2 | 0 | 0 | 0 | 0 |

**Effectiveness of Pre-training** As shown in Figure 1a and Figure 1b, benefiting from the pre-trained ViT, the training becomes more efficient, and most SSL algorithms achieve higher optimal performance. Note that Pseudo Labeling, Mean Teacher, Π model, VAT, and MixMatch barely converge if training WRN-28-8 from scratch. A possible reason is that the scarce labeled data cannot provide enough supervision for unlabeled data to form correct clusters. However, these methods can achieve sufficiently reasonable results when using pre-trained ViT. As illustrated in Figure 2, using ViT without pre-training performs the worst among different backbones. The reason can be that ViT is data hungry if trained from scratch [34, 72, 73]. However, after appropriate pre-training, ViT performs the best among all the backbones. In addition, we provide the T-SNE visualization of the features in Figure 3, where the pretrained ViT model demonstrates the most separable feature space after training. In a word, pre-trained ViT makes the training more efficient and improves the generalization performance of SSL algorithms. For NLP tasks, we observe similar results, yet the improvement can be relatively less significant since pre-training is the de-facto fashion in the field.

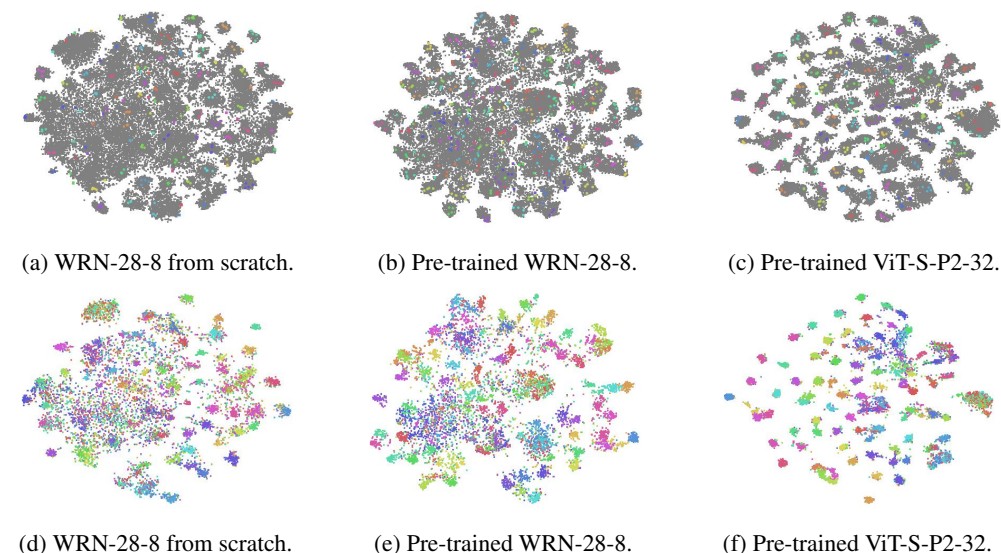

| (a) WRN-28-8 from scratch. | (b) Pre-trained WRN-28-8. | (c) Pre-trained ViT-S-P2-32. |
| (d) WRN-28-8 from scratch. | (e) Pre-trained WRN-28-8. | (f) Pre-trained ViT-S-P2-32. |

Figure 3: T-SNE visualization of FixMatch features on training data (first row) and testing data (second row) of CIFAR-100 (400 labels). Different colors refer to labeled data with different classes while unlabeled data is indicated by gray color.

**Robustness**   SSL sometimes hurts the generalization performance due to the large differences between the number of labeled data and the number of unlabeled data as shown in Table 9. We refer to an SSL algorithm as a robust SSL algorithm if it is consistently better than the supervised training setting. SSL algorithms cannot always outperform supervised training especially when labeled data is scarce. We find that CRMatch, AdaMatch and SimMatch are relatively robust SSL algorithms in USB. Although previous work has done some research towards robust SSL when using support vector machine [74, 75], we hope that our finding can serve as the motivation to delve into deep learning based robust SSL methods.

# 6   Codebase Structure of USB

In this section, we provide an overview of the codebase structure of USB, where four abstract layers are adopted. The layers include the core layer, algorithm layer, extension layer, and API layer in the bottom up direction as shown in Fig. 4.

**Core Layer**. In the core layer, we implement the commonly used core functions for training SSL algorithms. Besides, the code regarding datasets, data loaders, and models used in USB is also provided in the core layer. For flexible training, we implement common training hooks similar to MMCV [76], which can be modified and extended in the upper layers.

**Algorithm Layer**. In the algorithm layer, we first implement the base class for SSL algorithms, where we initialize the datasets, data loaders, and models from the core layer. Instead of implementing SSL algorithms independently as in TorchSSL [21], we further abstract the SSL algorithms, enabling better code reuse and making it easier to implement new algorithms. Except for the standalone implementation of loss functions used in SSL algorithms and algorithm-specific configurations, we further provide algorithm hooks according to the algorithm components summarized in Table 4. The algorithm hooks not only highlight the common part of different algorithms but also allows for a very easy and flexible combination of different components to resemble a new algorithm or conduct an ablation study. Based on this, we support 14 core SSL algorithms in USB, with two extra supervised learning variants. More algorithms are expected to be added through continued extension of USB.

**Extension Layer**. The extension layer is where we further extend the core SSL algorithms to different applications. Continuted effort are made on the extension of core SSL algorithms to imbalanced SSL algorithms [77, 78, 79, 80, 81, 82, 83, 84] and open-set SSL algorithms [85, 86, 87, 88, 89].

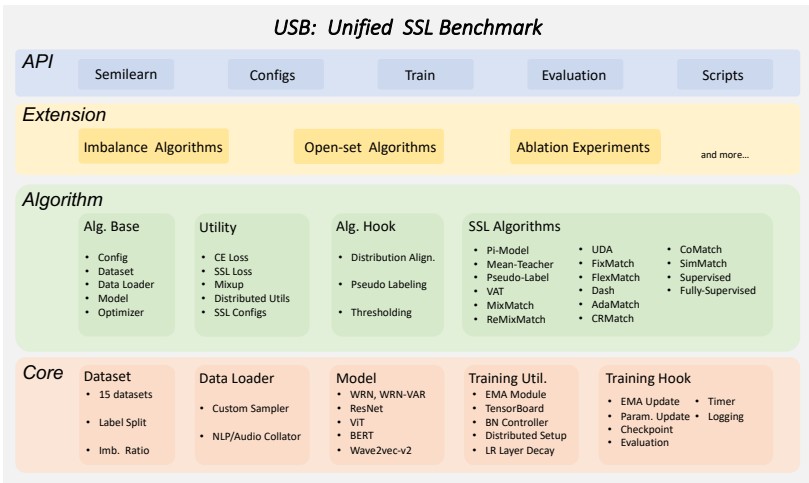

Figure 4: Structure of USB Codebase, consisting of 4 layers. The core layer provides the common functions, datasets, and models for SSL algorithms. The algorithm layer mainly implements the related SSL algorithms, with a high abstract level of algorithm components. Upon the algorithm layer, we use an extension layer for easy and flexible extension of core SSL algorithms. The top API layer supports a public python package SEMILEARN: `pip install semilearn`.

Systematic ablation study can also be conducted in the extension layer by inheriting either the core components and algorithms from the core layer or the algorithm layer.

**API Layer**. We wrap the core functions and algorithms in USB in the API layer as a public python package SEMILEARN. SEMILEARN is friendly for users from different backgrounds who want to employ SSL algorithms in new applications. Training and inference can be done in only a few lines of code with SEMILEARN. In addition, we provide the configuration files of all algorithms supported in USB with detailed parameter settings, which allows for reproduction of the results present in USB.

## 7 Limitation

Our primary focus is on semi-supervised classification in this paper. However, there are other SSL tasks that the SSL community should not ignore. USB currently does not include SSL tasks such as imbalanced semi-supervised learning [77, 79, 80, 81, 82, 83, 84], open-set semi-supervised learning [85, 86, 87, 88, 89], semi-supervised sequence modeling [90, 91, 92, 93, 26, 94], semi-supervised text generation [95, 96, 97], semi-supervised regression [98, 99, 100, 101, 102], semi-supervised object detection [103, 104, 105, 106, 107, 108], semi-supervised clustering [109, 110, 111, 112], etc. In addition, we do not implement generative adversarial networks based SSL algorithms [113, 64, 114, 65] and graph neural network based SSL algorithms [7, 115, 116, 117, 118] in USB, which are also important to the SSL community. Moreover, it is of great importance to extend current SSL to distributional shift settings, such as domain adaptation [119, 120] and out-of-distribution generalization [121], as well as time series anaysis [122]. We plan to evolve the benchmark in the future iterations over time by extending with more tasks.

## 8 Conclusion

We constructed USB, a unified SSL benchmark for classification that aims to enable consistent evaluation over multiple datasets from multiple domains and reduce the training cost to make the evaluation of SSL more affordable. With USB, we evaluate 14 SSL algorithms on 15 tasks across domains. We find that (1) although the performance of SSL algorithms is roughly close across domains, introducing diverse tasks from multiple domains is still necessary in the SSL scenario because the performance of SSL algorithms are not exactly steady across domains; (2) pre-training techniques can be helpful in the SSL scenario because it can not only accelerate the training but also improve the generalization performance; (3) unlabeled data sometimes hurts the performance especially when labeled data is extremely scarce. USB is a project for open extension and we plan to extend USB with more challenging tasks other than classification and introduce new algorithms.

## Acknowledgments

We would like to thank the anonymous reviewers for their insightful comments and suggestions to help improve the paper. The computing resources of this study were mainly supported by Microsoft Asia and partially supported by High-Flyer AI.

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

# A   Details of Datasets and in TorchSSL

We provide the details of datasets of TorchSSL in Table 10.

Table 10: Details of CV datasets and #labels used in TorchSSL. #Label per class represents the number of chosen labeled data per class from the training data. The test data is kept unchanged except for ImageNet where we use the validation dataset as the test dataset.

| Dataset | #Label per class | #Training data | #Test data | #Class |
|---------|------------------|----------------|------------|--------|
| CIFAR-10 | 4 / 25 / 100 | 50,000 | 10,000 | 10 |
| CIFAR-100 | 4 / 25 / 100 | 50,000 | 10,000 | 100 |
| SVHN | 4 / 25 /100 | 604,388 | 26,032 | 10 |
| STL-10 | 4 / 25 /100 | 100,000 | 10,000 | 10 |
| ImageNet | 100 | 1,281,167 | 50,000 | 1,000 |

# B   Correlation between TorchSSL and USB

Here we show the correlation between the mean error rates on TorchSSL and USB CV tasks. We take the 14 algorithms considered in the main paper and show their mean performance on TorchSSL versus that on USB CV tasks in Figure 5. Despite the fact that the Pearson correlation coefficient is 0.85, the final rank SSL algorithms is not consistent, which shows different adaptability of different methods when using pre-trained ViTs. For example, ReMixMatch shows the best mean performance on USB while AdaMatch and SimMatch have the best mean performance on TorchSSL. Please refer to Table 8 for more detailed rankings on CV, NLP, and Audio.

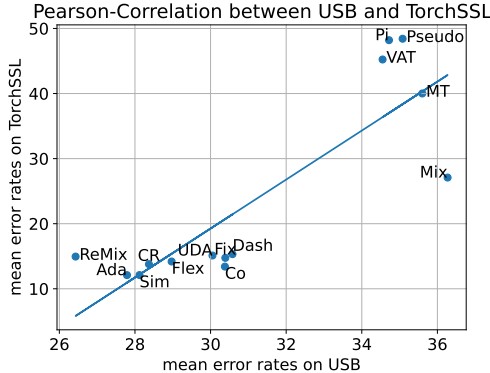

Figure 5: Correlation between TorchSSL and USB.

# C   Performance Results on ImageNet

Although we have excluded ImageNet from USB, we provide an evaluation on ImageNet of MAE pre-trained ViT-B, using UDA [29], FixMatch [20], FlexMatch [21], CoMatch [60], and SimMatch [47]. We train these algorithms using 10 labels per class and 100 labels per-class, i.e., a total of 10,000 labels and 100,000 labels respectively, corresponding to roughly 1% and 10% of the total labeled data in ImageNet. For learning rate and weight decay, we follow the fine-tuning protocol in MAE [33], where we use AdamW with a learning rate of 1e-3 and weight decay of 0.05. We use 16 A100 to train each algorithm and set the batch size to 256 for both labeled and unlabeled data. Other algorithmic hyper-parameters stay the same as their original implementations.

We present the results on ImageNet in Table 11. UDA and Fixmatch are near the bottom, similar to USB. SimMatch is still marked as one of the tops. Surprisingly, CoMatch does so well on ImageNet when it ranked only 9th on the USB benchmark. Also, while FlexMatch is the best on USB, it's pretty firmly behind CoMatch and SimMatch on ImageNet.

Table 11: ImageNet **accuracy** results. We use MAE pre-trained ViT-B.

| Method | 1w Labels | 10w Labels | Rank |
|---|---|---|---|
| UDA | 38.62 | 62.37 | 5 |
| FixMatch | 37.93 | 62.88 | 4 |
| FlexMatch | 39.13 | 63.09 | 3 |
| CoMatch | 44.32 | 65.80 | 2 |
| SimMatch | 46.48 | 67.61 | 1 |

Table 12: Swin-Transformer results on EuroSAT and Semi-AVES.

| Dataset | EuroSAT | | Semi-Aves |
|---|---|---|---|
| # Label | 20 | 40 | 5,959 |
| Supervised | $44.32_{\pm1.10}$ | $34.40_{\pm1.44}$ | $38.76_{\pm0.21}$ |
| Fully-Supervised | $1.86_{\pm0.10}$ | | - |
| Π-Model | $42.49_{\pm3.21}$ | $30.54_{\pm1.37}$ | $38.74_{\pm0.60}$ |
| Pseudo-Labeling | $42.49_{\pm3.21}$ | $30.54_{\pm1.37}$ | $38.74_{\pm0.60}$ |
| Mean Teacher | $35.85_{\pm1.95}$ | $19.62_{\pm3.28}$ | $33.37_{\pm0.06}$ |
| VAT | $40.63_{\pm2.68}$ | $29.94_{\pm1.87}$ | $35.84_{\pm0.36}$ |
| UDA | $18.15_{\pm5.70}$ | $12.09_{\pm1.26}$ | $29.28_{\pm0.20}$ |
| FixMatch | $17.19_{\pm3.46}$ | $12.57_{\pm1.28}$ | $28.88_{\pm0.22}$ |
| Dash | $18.04_{\pm1.21}$ | $12.98_{\pm1.27}$ | $28.69_{\pm0.39}$ |
| CoMatch | $13.65_{\pm1.42}$ | $10.17_{\pm0.68}$ | $37.71_{\pm0.31}$ |
| CRMatch | $30.28_{\pm1.64}$ | $22.39_{\pm1.41}$ | $29.22_{\pm0.21}$ |
| FlexMatch | $10.46_{\pm1.20}$ | $9.06_{\pm1.80}$ | $30.19_{\pm0.51}$ |
| SimMatch | $11.19_{\pm1.01}$ | $10.65_{\pm1.64}$ | $28.55_{\pm0.13}$ |

# D    Results with Different Pre-trained Backbones

In this section, we verify USB with different pre-trained backbones. Different pre-trained backbones do affect the performance of SSL algorithms, which makes it important to report results with multiple backbones. We will continuously update results with different backbones at `https://github.com/microsoft/Semi-supervised-learning`. Here we report several results in Table 12, Table 13, and Table 14. Across the tasks, there is a pretty clear distinction between the performance of algorithms in the first half of the ranking list and the second half of the ranking list. While switching out backbones does not change the membership of these two halves, it does seem like the relative orderings within the top half can indeed vary a bit.

To compare different backbones on CV tasks, we fine-tune pre-trained public Swin-Transformer [123] with USB. We keep all hyper-parameters the same as in Table 15, and mainly evaluate on EuroSAT (32) and Semi-Aves (224). For EuroSAT, we change the input image size of the pre-trained Swin-S from 224 to 32, and the window size from 7 to 4 to accommodate the adapted input image size. For Semi-Aves, we adopt the original Swin-S. From the results in Table 12, one can observe, that on EuroSAT (32), as we adopt 224 pre-trained Swin-S and change its input and window size, the results are inferior to ViT-32 reported in the paper, whereas on Semi-Aves (224), the results are better than ViT-S. An interesting finding is that CoMatch performs relatively better with Swin-S while CrMatch performs worse. This also shows the importance of constantly updating the backbone in the future development of USB.

For NLP tasks, we additionally experiment with RoBERTa [31]. We train RoBERTa using the same hyper-parameters reported in Table 16. RoBerta generally performs better than Bert as expected. The performance difference is both very close when using RoBerta or Bert.

Due to the fact that the audio tasks setting in the current version of USB being built upon raw waveforms, there are not many pre-trained models available to use. We report the results of HuBert [32] and Wave2Vecv2.0 [71] for audio tasks to compare different backbones. The difference between these two backbones selected mainly lies in pre-training data. Wave2Vecv2.0 is pre-trained using raw human voice data and HuBert is an improved model with a discrete clustering target. Thus we can observe from the results, that on human voice tasks Superb-KS, Wave2Vecv2.0 has better performance, whereas, on other tasks, HuBert is more robust and outperforms Wave2Vecv2.0.

Table 13: RoBERTa results on Yelp.

| Dataset | Yelp | |
|---|---|---|
| # Labels | 250 | 1000 |
| Supervised | $42.56_{\pm1.15}$ | $39.00_{\pm0.16}$ |
| Fully-Supervised | $29.15_{\pm0.12}$ | |
| Pseudo-Label | $48.26_{\pm0.02}$ | $40.56_{\pm0.16}$ |
| MeanTeacher | $49.41_{\pm0.03}$ | $44.36_{\pm1.04}$ |
| Π-Model | $49.16_{\pm2.04}$ | $42.93_{\pm0.88}$ |
| VAT | $43.04_{\pm0.02}$ | $39.24_{\pm0.06}$ |
| AdaMatch | $38.24_{\pm0.02}$ | $35.64_{\pm0.06}$ |
| UDA | $40.13_{\pm0.15}$ | $38.98_{\pm0.03}$ |
| FixMatch | $39.82_{\pm0.95}$ | $37.42_{\pm0.30}$ |
| FlexMatch | $39.11_{\pm0.02}$ | $36.84_{\pm0.01}$ |
| Dash | $39.86_{\pm1.01}$ | $36.23_{\pm0.21}$ |
| CRMatch | $40.08_{\pm1.28}$ | $35.85_{\pm0.38}$ |
| CoMatch | $39.95_{\pm0.86}$ | $36.89_{\pm0.22}$ |
| SimMatch | $38.76_{\pm0.68}$ | $36.39_{\pm0.34}$ |

Table 14: HuBert results on keyword Spotting and Wave2Vec2.0 results on FSDnoisy.

| Dataset | keyword Spotting | | FSDnoisy |
|---|---|---|---|
| # Label | 50 | 400 | 1,772 |
| Supervised | $8.95_{\pm1.62}$ | $6.31_{\pm0.46}$ | $33.54_{\pm1.65}$ |
| Fully-Supervised | $2.41_{\pm0.15}$ | | - |
| Π-Model | $87.86_{\pm2.88}$ | $72.89_{\pm3.23}$ | $35.97_{\pm0.84}$ |
| Pseudo-Labeling | $25.59_{\pm2.88}$ | $13.02_{\pm2.47}$ | $35.23_{\pm0.78}$ |
| Mean Teacher | $89.79_{\pm0.30}$ | $90.01_{\pm0.02}$ | $40.13_{\pm1.70}$ |
| VAT | $2.27_{\pm0.07}$ | $2.43_{\pm0.02}$ | $34.21_{\pm0.31}$ |
| UDA | $11.76_{\pm0.06}$ | $2.23_{\pm0.16}$ | $33.09_{\pm1.03}$ |
| FixMatch | $11.63_{\pm0.24}$ | $8.93_{\pm2.04}$ | $33.09_{\pm0.64}$ |
| Dash | $11.88_{\pm0.15}$ | $8.25_{\pm4.22}$ | $33.02_{\pm1.39}$ |
| CoMatch | $15.96_{\pm1.02}$ | $10.34_{\pm1.52}$ | $30.24_{\pm0.55}$ |
| CRMatch | $5.85_{\pm1.19}$ | $3.66_{\pm0.33}$ | $30.48_{\pm0.65}$ |
| FlexMatch | $10.22_{\pm1.10}$ | $5.10_{\pm3.70}$ | $32.66_{\pm4.09}$ |
| SimMatch | $9.43_{\pm0.63}$ | $5.47_{\pm2.72}$ | $29.57_{\pm0.52}$ |

# E   Details of Datasets in USB

## E.1   CV Tasks

**CIFAR-100**   The CIFAR-100 [39] dataset is a natural image ($32\times32$ pixels) recognition dataset consisting 100 classes. There are 500 training samples and 100 test samples per class.

**STL-10**   The STL-10 [40] dataset is a natural color image ($96\times96$ pixels) recognition dataset consisting 10 classes. Particularly, each class has 500 training samples and 800 test samples. Apart from the labeled samples, STL-10 also provides 100,000 unlabeled samples. Note that the unlabeled samples contain other classes in addition to the ones in the labeled data.

**EuroSat**   EuroSAT [43, 44] dataset is based on Sentinel-2 satellite images covering 13 spectral bands and consisting of 10 classes with 27,000 labeled and geo-referenced samples. Following [124], we use the dataset with the optical R, G, B frequency bands, thus each image is of size $64 \times 64 \times 3$. We take the first 60% images from each class as training set; the next 20% as val set, and the last 20% as test set.

**TissueMNIST**   TissueMNIST [41, 42] is a medical dataset of human kidney cortex cells, segmented from 3 reference tissue specimens and organized into 8 categories. The total 236,386 training samples are split with a ratio of 7 : 1 : 2 into training (165,466 images), validation (23,640 images) and test set (47,280 images). Each gray-scale image is $28 \times 28$ pixels.

**Semi-Aves**    Semi-Aves [45] is a dataset of Aves (birds) classification, where 5,959 images of 200 bird species are labeled and 26,640 images are unlabeled. As class distribution mismatch hurts the performance [85], we do not use out-of-class unlabeled data. This dataset is challenging as it is naturally imbalanced. The validation and test set contain 10 and 20 images respectively for each of the 200 categories in the labeled set.

## E.2    NLP Tasks

**IMDB**    The IMDB [49] dataset is a binary sentiment classification dataset. There are 25,000 reviews for training and 25,000 for test. IMDB is class balanced which means the positive and negative reviews have the same number both for training and test. For USB, we draw 12,500 samples and 1,000 samples per class from training samples to form the training dataset and validation dataset respectively. The test dataset is unchanged.

**Amazon Review**    The Amazon Review [52] dataset is a sentiment classification dataset. There are 5 classes (scores). Each class (score) contains 600,000 training samples and 130,000 test samples. For USB, we draw 50,000 samples and 5,000 samples per class from training samples to form the training dataset and validation dataset respectively. The test dataset is unchanged.

**Yelp Review**    The Yelp Review [53] sentiment classification dataset has 5 classes (scores). Each class (score) contains 130,000 training samples and 10,000 test samples. For USB, we draw 50,000 samples and 5,000 samples per class from training samples to form the training dataset and validation dataset respectively. The test dataset is unchanged.

**AG News**    The AG News [50] dataset is a news topic classification dataset containing 4 classes. Each class contains 30,000 training samples and 1,900 test samples. For USB, we draw 25,000 samples and 2,500 samples per class from training samples to form the training dataset and validation dataset respectively. The test dataset is unchanged.

**Yahoo! Answer**    The Yahoo! Answer [51] topic classification dataset has 10 categories. Each class contains 140,000 training samples and 6,000 test samples. For USB, we draw 50,000 samples and 5,000 samples per class from training samples to form the training dataset and validation dataset respectively. The test dataset is unchanged.

## E.3    Audio Tasks

**GTZAN**    The GTZAN dataset is collected for music genre classification of 10 classes and 100 audio recordings for each class. The maximum length of the recordings is 30 seconds and the original sampling rate is 22,100 Hz. We split 7,000 samples for training, 1,500 for validation, and 1,500 for testing. All recordings are re-sampled at 16,000 Hz.

**UrbanSound8k**    The UrbanSound8k dataset [54] contains 8,732 labeled sound events of urban sounds of 10 classes, with the maximum length of 4 seconds. The original sampling rate of the audio recordings is 44,100 and we re-sample it to 16,000. It is originally divided into 10 folds, where we use the first 8 folds of 7,079 samples as training set, and the last two folds as validation set of size 816 and testing set of size 837 respectively.

**FSDNoisy18k**    The FSDNoisy18 dataset [56] is a sound event classification dataset across 20 classes. It consists of a small amount of manually labeled data - 1,772 and a large amount of noisy data - 15,813 which is treated as unlabeled data in our paper. The original sample rate is 44,100 Hz, and the length of the recordings lies between 3 seconds and 30 seconds. We use the testing set provided for evaluation, which contains 947 samples.

**Keyword Spotting (Superb-KS)**    The Keyword spotting dataset is one of the tasks in Superb [57] for classifying the keywords. It contains speech utterances of a maximum length of 1 second and the sampling rate of 16,000. The training, validation, and testing set contain 18,538; 2,577; 2,567 recordings, respectively. For pre-processing, we remove the silence and unknown labels from the dataset.

**ESC-50**   The ESC-50 [55] is a dataset containing 2,000 environmental audio recordings for 50 sound classes. The maximum length of the recordings is 5 seconds and the original sampling rate is 44,100. We split 1,200 samples as training data, 400 as validation data, and 400 as testing data. We also re-sample the audio recordings to 16,000 Hz during pre-processing.

# F   Details of Implemented SSL algorithms in USB

Π **model [35]** is a simple SSL algorithm that forces the output probability of perturbed versions of unlabeled data be the same. Π model uses Mean Squared Error (MSE) for optimization.

**Pseudo Labeling [59]** turns the output probability of unlabeled data into the 'one-hot' hard one and makes the same unlabeled data to learn the pseudo 'one-hot' label. Unlike Π model, Pseudo Labeling uses CE for optimization.

**Mean Teacher [36]** takes the exponential moving average (EMA) of the neural model as the teacher model. With Mean Teacher, the neural model forces itself to output a similar probability to the EMA teacher. Though the later SSL algorithms will not always choose the EMA model as the teacher, they often use the EMA model for validation/test cause it decreases the risk of neural models falling into the local optima.

**VAT [37]** enhances the robustness of the conditional predicted label distribution around each unlabeled data against an adversarial perturbation. In other words, VAT forces the neural model to give similar predictions on unlabeled data even facing a strong adversarial perturbation.

**MixMatch [28]** first introduces Mixup [125] into SSL by taking the input as the mixture of labeled and unlabeled data and the output as the mixture of labels and model predictions on unlabeled data. Note that MixMatch also utilizes MSE as the unsupervised loss.

**ReMixMatch [23]** can be seen as the upgraded version of MixMatch. ReMixMatch improves MixMatch by (1) proposing stronger augmentation (i.e., Control Theory Augmentation (CTAugment) [23]) for unlabeled data; (2) using Augmentation Anchoring to force the model to output similar predictions to weakly augmented unlabeled data when fed strongly augmented data; (3) utilizing Distribution Alignment to encourage the marginal distribution of predictions on unlabeled data to be similar to the marginal distribution of labeled data.

**UDA [29]** also introduces strong augmentation (i.e., RandAugment [126]) for unlabeled data. The core idea of UDA is similar to Augmentation Anchoring [23], which forces the predictions of neural models on the strongly-augmented unlabeled data to be close to those of weakly-augmented unlabeled data. Instead of turning predictions into hard 'one-hot' pseudo-labels, UDA sharpens the prediction on unlabeled data. Thresholding technique is used to mask out unconfident unlabeled samples that are considered noise here.

**FixMatch [20]** is the upgraded version of Pseudo Labeling. FixMatch turns the predictions on weakly-augmented unlabeled data into hard 'one-hot' pseudo-labels and then further uses them as the learning signal of strongly-augmented unlabeled data. FixMatch finds that using a high threshold (e.g., 0.95) to filter noisy unlabeled predictions and take the rest as the pseudo-label can achieve very good performance.

**Dash [24]** improves the FixMatch by using a gradually increased threshold instead of a fixed threshold, which allows more unlabeled data to participate in the training at the early stage. Moreover, Dash theoretically establishes the convergence rate from the view of non-convex optimization.

**CoMatch [60]** firstly introduces contrastive learning into SSL. Except for consistency regularizing on the class probabilities, it is also exploited on graph-based feature representations, which impose smooth constraints on pseudo-labels generated.

**CRMatch [61]** proposed an improved consistency regularization framework which impose consistency and equivariance on the classification probability and the feature level.

**FlexMatch [21]** firstly introduces the class-specific thresholds into SSL by considering the different learning difficulties of different classes. Specifically, the hard-to-learn classes should have a low threshold to speed up convergence while the easy-to-learn classes should have a high threshold to avoid confirmation bias.

**AdaMatch [62]** is proposed mainly for domain adaption, but can also adapted to SSL. It is characterized by Relative Threshold and Distribution Alignment, where the relative threshold is adaptively estimated from EMA of the confidence on labeled data.

**SimMatch [47]** extends CoMatch [60] by considering semantic-level and instance-level consistency regularization. Similar similarity relationship of different augmented versions on the same data with respect to other instances is encouraged during training. In addition, a memory buffer consisting of predictions on labeled data is adopted to connect the two-level regularization.

# G   Experiment Details in USB

## G.1   Setup for CV Tasks in USB

Table 15: Hyper-parameters of CV tasks in USB.

| Dataset | CIFAR-100 | STL-10 | Euro-SAT | TissueMNIST | Semi-Aves |
|---|---|---|---|---|---|
| Image Size | 32 | 96 | 32 | 32 | 224 |
| Model | ViT-S-P4-32 | ViT-B-P16-96 | ViT-S-P4-32 | ViT-T-P4-32 | ViT-S-P16-224 |
| Weight Decay | | | 5e-4 | | |
| Labeled Batch size | | | 16 | | |
| Unlabeled Batch size | | | 16 | | |
| Learning Rate | 5e-4 | 1e-4 | 5e-5 | 5e-5 | 1e-3 |
| Layer Decay Rate | 0.5 | 0.95 | 1.0 | 0.95 | 0.65 |
| Scheduler | | | $\eta = \eta_0 \cos(\frac{7\pi k}{16K})$ | | |
| Model EMA Momentum | | | 0.0 | | |
| Prediction EMA Momentum | | | 0.999 | | |
| Weak Augmentation | | | Random Crop, Random Horizontal Flip | | |
| Strong Augmentation | | | RandAugment [126] | | |

For CV tasks in USB, we use ViT models [34]. We find that directly using released ViT models leads to overfitting and one needs to fix the image resolution as the pre-trained resolution, as demonstrated in Paragraph 5.4. Instead, we pre-train our own ViT models on ImageNet-1K [8]. To match the number of parameters as the CNN models used in the classic setting, we use ViT-Tiny and ViT-Small with a patch size of 2 and image size of 32 for TissueMNIST, CIFAR-100 and EuraSAT, respectively; ViT-Small with a patch size of 16 and image size of 224 for Semi-Aves. For better transfer performance, we adopt an MLP before the final classifier during pre-training, as in [127]. For supervised pre-training on ImageNet-1K, we use Lamb optimizer with a learning rate of 0.05, and a weight decay of 0.03 for ViT-Tiny and a weight decay of 0.05 for ViT-Small. We adopt a large batch size of 4096 and train the networks for 300 epochs, with a linear learning rate warmup for the first 20 epochs. After the warmup, cosine scheduler is utilized. For augmentation, we use RandAugment [126], along with Mixup [125] and CutMix [128]. We also use label smoothing of 0.1 during pre-training. Since STL10 is a subset of ImageNet, we adopt unsupervised pre-training MAE [33] of ViT-Small with image size of 96 to avoid cheating.

For USB CV tasks, we adopt layer-wise learning rate decay as in [123]. We tune the learning rate and layer decay rate on different datasets using FixMatch, and use the best configuration to train all SSL algorithms [7]. The cosine annealing scheduler is similar to the classic setting but with total steps of $204, 800$ and a warm-up of $5, 120$ steps. The labeled and unlabeled batch size is both set to 16. Other algorithm-related hyper-parameters stay the same as in the original papers.

## G.2   Setup for NLP Tasks in USB

We use pre-trained BERT-Base [30] for all NLP tasks in USB. We set the batch size of labeled data and unlabeled data to 4 for reducing the training time and GPU memory requirement. To fine-tune the BERT-Base under USB, we adopt AdamW optimizer with weight decay of $1e{-}4$. Similarly, we conduct a grid search of the learning rate and layer decay on different datasets using FixMatch and pick the best configuration to fine-tune other SSL algorithms. We utilize the same cosine learning rate

---

[7]We present the full tuning results in: `https://github.com/microsoft/Semi-supervised-learning`.

scheduler as in the classic setting with the total training steps of $102,400$ and a warm-up of $5,120$ steps. We use the fine-tuned model without parameter momentum to conduct evaluations. For all datasets, we cut the long sentence to satisfy the input length requirement of BERT-Base. For data augmentation, we adopt back-translation as the strong augmentation [29, 25]. Specifically, we use De-En and Ru-En translation with WMT19.

Table 16: Hyper-parameters of NLP tasks in USB.

| Dataset | AG News | Yahoo! Answer | IMDb | Amazom-5 | Yelp-5 |
|---|---|---|---|---|---|
| Max Length | | | 512 | | |
| Model | | | Bert-Base | | |
| Weight Decay | | | 1e-4 | | |
| Labeled Batch size | | | 4 | | |
| Unlabeled Batch size | | | 4 | | |
| Learning Rate | 5e-5 | 1e-4 | 5e-5 | 1e-5 | 5e-5 |
| Layer Decay Rate | 0.65 | 0.65 | 0.75 | 0.75 | 0.75 |
| Scheduler | | | $\eta = \eta_0 \cos(\frac{7\pi k}{16K})$ | | |
| Model EMA Momentum | | | 0.0 | | |
| Prediction EMA Momentum | | | 0.999 | | |
| Weak Augmentation | | | None | | |
| Strong Augmentation | | | Back-Translation [29] | | |

## G.3 Setup for Audio Tasks in USB

For Audio tasks, we adopt Wav2Vec 2.0 [71] and HuBert [32] as the pre-trained model. The batch size of labeled data and unlabeled data is set to $8$. We keep the sampling rate of audios as $16,000$. We adopt AdamW optimizer with a weight decay of $5e{-}4$, and search the learning rate and layer decay as before. Other hyper-parameter settings are the same as NLP tasks. Mimicking RandAugment, for strong augmentation in audio tasks, we random sample 2 augmentations from the augmentation pool and random set the augmentation magnitude during training.

Table 17: Hyper-parameters of Audio tasks in USB.

| Dataset | GTZAN | Keyword Spotting | UrbanSound8k | FSDNoisy | ESC-50 |
|---|---|---|---|---|---|
| Sampling Rate | | | 16,000 | | |
| Max Length | 3.0 | 1.0 | 4.0 | 5.0 | 5.0 |
| Model | Wav2Vecv2-Base | | HuBERT-Base | | |
| Weight Decay | | | 5e-4 | | |
| Labeled Batch size | | | 8 | | |
| Unlabeled Batch size | | | 8 | | |
| Learning Rate | 2e-5 | 5e-5 | 5e-5 | 5e-4 | 1e-4 |
| Layer Decay Rate | 1.0 | 0.75 | 0.75 | 0.75 | 0.85 |
| Scheduler | | | $\eta = \eta_0 \cos(\frac{7\pi k}{16K})$ | | |
| Model EMA Momentum | | | 0.0 | | |
| Prediction EMA Momentum | | | 0.999 | | |
| Weak Augmentation | | | Random Sub-sample | | |
| Strong Augmentation | | Random Sub-sample, Random Gain, Random Pitch, Random Speed | | | |

