# OpenReview forum: "USB: A Unified Semi-supervised Learning Benchmark for Classification"
_NeurIPS.cc/2022/Track/Datasets_and_Benchmarks — NeurIPS 2022 Datasets and Benchmarks _

### Official Review · Reviewer_YTad · 2022-06-30
**Paper is well-motivated. The proposed benchmark is of high importance in developing SSL algorithms. However, more clarifications and algorithms are required.**

**Rating:** 7
**Confidence:** 3
**Clarity:** The paper is, in general, well-written.

**Strengths:**

1- The paper, in general, is well-motivated specially as it proposes to facilitate the development of SSl algorithms to research labs with limited computational resources.

2- The variety of the datasets across multiple domains, and the considered 14 SSL algorithms.


**Weaknesses:**

1- In general, further clarifications are needed for some parts of the paper (points 2 and 3 below).

2- For the sentence "Compared with training neural models from scratch, pre-training has much reduced cost in SSL, yet relatively few benchmarks offer a fair test bed for SSL with the pre-trained versions of neural models.", can authors provide references? If there are other test beds for SSL methods, it is recommended to mention them and exploit their limitations by which USB is addressing in this paper. I understand that the authors show that difference in GPU days needed to evaluate SSL algorithms in Table 1a with referencing [21]. However, further explanations and details into the weaknesses of [21] would improve the paper.

3- No structural details and layers stacks are provided for the utilized ViT. Given that the main contribution of this work is the use of pre-trained ViT as an evaluation benchmark, I believe describing how the ViT is utilized along with its structure is of high importance. Also, it is not clear, for instance, if the same architecture is used for all tasks. It is mentioned that for CV, authors use a model similar to [27]. For NLP and Audio, on the other hand, it is mentioned that a transformer model similar to [64-66] is adopted. Does this mean that for each task, a different pre-trained model is used?

4- While the Limitation section is appreciated, it is not clear why none of the mentioned methods in this section is implemented. A justification is needed in this case which leads to the question: Can USB be considered unified if popular GAN-based and/or GCN-based SSL methods are not implemented?


**Additional Feedback:**

1- Not all SSl algorithms achieve optimal results on Pre-Trained ViT-S-P2-32 (Figure 1b). Can this observation be justified?

2- Run-time results for Tables 5-9.


**Correctness:**

The claims made are mostly correct. The proposed evaluation method can be strengthened if GAN-based and/or GCN-based methods are included. This would be more valuable when compared to using similar methods such as MiXMatch, ReMixMatch, and FixMatch.

**Documentation:**

Yes.

**Ethics:**

No ethical concerns.

**Relation To Prior Work:**

The authors mention plenty of previous SSL works. However, further details are needed to compare against existing evolution protocols in reference [21]. See the weaknesses section.

**Summary And Contributions:**

Existing SSL methods evaluation protocols are limited because of (i) only implementing them using CV tasks, and (ii) the requirement to train deep neural models from scratch. This paper evaluates different SSL algorithms across different domains (CV, NLP, and Audio Tasks) using vision Transformers (ViT) instead of training conventional structures such as ResNet from scratch. The main contribution is providing the pre-trained models that serve as a benchmark for SSL algorithms. These evaluation models are based on ViT.

---

> ### Author Response · Authors · 2022-08-17
> **Response to Reviewer YTad - Part1**
>
> **Weaknesses1:**  For the sentence "Compared with training neural models from scratch, pre-training has much reduced cost in SSL, yet relatively few benchmarks offer a fair test bed for SSL with the pre-trained versions of neural models.", can authors provide references? If there are other test beds for SSL methods, it is recommended to mention them and exploit their limitations by which USB is addressing in this paper. I understand that the authors show that difference in GPU days needed to evaluate SSL algorithms in Table 1a with referencing [21]. However, further explanations and details into the weaknesses of [21] would improve the paper.
>
> **Answer:**  The weaknesses of TorchSSL [21] are
>
> (1) It can be time-consuming and environmentally unfriendly because it typically trains deep neural models from scratch.
>
> (2) It is constrained to plain computer vision (CV) tasks precluding consistent and diverse evaluation over tasks in natural language processing (NLP), and audio processing (Audio). To our humble knowledge, we are the first to discuss whether current SSL methods that work well on CV tasks still work on NLP and Audio tasks.
>
> We have made it clear in the revised paper.
>
> **Weaknesses2:** No structural details and layers stacks are provided for the utilized ViT. Given that the main contribution of this work is the use of pre-trained ViT as an evaluation benchmark, I believe describing how the ViT is utilized along with its structure is of high importance. Also, it is not clear, for instance, if the same architecture is used for all tasks. It is mentioned that for CV, authors use a model similar to [27]. For NLP and Audio, on the other hand, it is mentioned that a transformer model similar to [64-66] is adopted. Does this mean that for each task, a different pre-trained model is used?
>
> **Answer:**  This part of the information is actually included in our Appendix and we have revised this part to make it more clear. We find that directly using pre-trained ViT of image size 224 and patch size 16 leads to overfitting on CIFAR100, SVHN, CIFAR10 with an image size of 32. We think this is related to pre-training image size and too-large patch size for public pre-trained ViT. One way to resolve this is to up-sample images to 224, but this will increase many computations which violate our motivation. Thus we trained our own ViTs on ImageNet with image size of 32 and patch size of 4, e.g. vit_small_patch4_32 (23M) and vit_tiny_patch4_32 (6M). The exact ViT used for each task is selected to match the parameter size of the original backbones used in the classic setting, e.g. vit_tiny_patch4_32 replaces WRN-28-2 for CIFAR10 and SVHN and vit_small_patch4_32 replaces WRN-28-8 for CIFAR100. The exact model used for each dataset is indicated in Appendix.
>
> **Weaknesses3:** While the Limitation section is appreciated, it is not clear why none of the mentioned methods in this section is implemented. A justification is needed in this case which leads to the question: Can USB be considered unified if popular GAN-based and/or GCN-based SSL methods are not implemented?
>
> **Answer:** Thanks for the question. We did not include GCN-based methods because their evaluation is on graph/node classification tasks, which are radically different from image classification tasks and SSL algorithms except GCN-based methods are also not suitable for graph/node classification tasks. We did not include GAN-based methods because their performance on the existing benchmark is inferior to most consistency-based methods (R3-CGAN has a test error rate of 6.69% on CIFAR-10 with 4000 labels while FixMatch in the same year achieves 4.21%) and applying pre-trained ViT on them is not trivial and often leads to unstable training. That being said, we will tone down and repitch the paper accordingly and will consider making a graph/node classification benchmark for semi-supervised learning in the future.

---

> > ### Comment · Reviewer_YTad · 2022-08-22
> > **Authors addressed all main concerns**
> >
> > The authors have addressed most of main concerns. Given their response and the changes made, I will increase the score.

---

> ### Author Response · Authors · 2022-08-17
> **Response to Reviewer YTad - Part2**
>
> **Correctness:** The claims made are mostly correct. The proposed evaluation method can be strengthened if GAN-based and/or GCN-based methods are included. This would be more valuable when compared to using similar methods such as MiXMatch, ReMixMatch, and FixMatch.
>
> **Answer:** Thanks for the suggestion. As argued above, GCN-based methods are not suitable for comparison in the same benchmark as other SSL methods. As for GAN-based methods, we provide results of MarginGAN, which is the most recent GAN-based method with code available, using the TorchSSL benchmark in the table below:
>
> |                  | CIFAR10-40 Labels         | CIFAR10-250 Labels        | CIFAR10-4000 Labels       |   | CIFAR100-400 Labels        | CIFAR100-2500 Labels       | CIFAR100-10000 Labels      |
> |------------------|------------|------------|------------|---|------------|------------|------------|
> | FullySupervised  | 95.38±0.05 | 95.39±0.04 | 95.38±0.05 |   | 80.70±0.09 | 80.70±0.09 | 80.73±0.05 |
> | PiModel          | 25.66±1.76 | 53.76±1.29 | 86.87±0.59 |   | 13.04±0.80 | 41.20±0.66 | 63.35±0.00 |
> | PseudoLabel      | 25.39±0.26 | 53.51±2.20 | 84.92±0.19 |   | 12.55±0.85 | 42.26±0.28 | 63.45±0.24 |
> | PseudoLabel_Flex | 26.26±1.96 | 53.86±1.81 | 85.25±0.19 |   | 14.28±0.46 | 43.88±0.51 | 64.40±0.15 |
> | MeanTeacher      | 29.91±1.60 | 62.54±3.30 | 91.90±0.21 |   | 18.89±1.44 | 54.83±1.06 | 68.25±0.23 |
> | VAT              | 25.34±2.12 | 58.97±1.79 | 89.49±0.12 |   | 14.80±1.40 | 53.16±0.79 | 67.86±0.19 |
> | MixMatch         | 63.81±6.48 | 86.37±0.59 | 93.34±0.26 |   | 32.41±0.66 | 60.24±0.48 | 72.22±0.29 |
> | ReMixMatch       | 90.12±1.03 | 93.70±0.05 | 95.16±0.01 |   | 57.25±1.05 | 73.97±0.35 | 79.98±0.27 |
> | UDA              | 89.38±3.75 | 94.84±0.06 | 95.71±0.07 |   | 53.61±1.59 | 72.27±0.21 | 77.51±0.23 |
> | UDA_Flex         | 94.56±0.52 | 94.98±0.07 | 95.76±0.06 |   | 54.83±1.88 | 72.92±0.15 | 78.09±0.10 |
> | FixMatch         | 92.53±0.28 | 95.14±0.05 | 95.79±0.08 |   | 53.58±0.82 | 71.97±0.16 | 77.80±0.12 |
> | FlexMatch        | 95.03±0.06 | 95.02±0.09 | 95.81±0.01 |   | 60.06±1.62 | 73.51±0.20 | 78.10±0.15 |
> | MarginGAN        | 17.78±1.94 | 53.02±11.36| 92.87±0.12 |   | 13.70±1.80 | 48.01±0.46 | 69.54±0.31 |
>
>
> MarginGAN's performance at a low data regime (40 labels on CIFAR10 and 400 labels on CIFAR-100) is much lower than that of consistency-based methods.
>
>
> **Relation To Prior Work:**
> The authors mention plenty of previous SSL works. However, further details are needed to compare against existing evolution protocols in reference [21]. See the weaknesses section.
>
> **Answer:** See the Answer for weaknesses 1.
>
> **Additional Feedback1:** Not all SSL algorithms achieve optimal results on Pre-Trained ViT-S-P2-32 (Figure 1b). Can this observation be justified?
>
> **Answer:**  Figure 1b just illustrates the convergence results before 100k steps while we trained each algorithm for 200k steps. We did not plot the latter 100k steps since almost all SSL algorithms change slightly.
>
> **Additional Feedback2:** Run-time results for Tables 5-9.
>
> **Answer:**  The detailed run-time results are plotted in Table 1.

---

### Official Review · Reviewer_ch9E · 2022-07-05
**A comprehensive SSL benchmarking on vision, NLP and audio**

**Rating:** 7
**Confidence:** 3
**Correctness:** Overall well-written.
**Clarity:** All is clear.

**Strengths:**

1. This paper builds a comprehensive benchmarking on Semi-supervised Learning, which contains vision, NLP and audio.

2. The work opens the source code for further extension, which is beneficial for the ML community in the SSL field.

3. This paper is well-written and provides details on the datasets and experiment setup.

4. It is great that "We plan to evolve the benchmark in the future iterations over time by extending with more tasks." (Line 220-221) This benchmarking could be impactful for the SSL community.


**Weaknesses:**

1. Although the experiment results are comprehensive, the insights and analysis are relatively limited. It would be better if the authors could provide more insights and discoveries from the experiment results. These insights are critical for inspiring SSL model designs in the future.

2. It is suggested that what the benchmark can help for the SSL community could be described more clearly and explicitly in the paper.

3. Since there are already a lot of existing SSL benchmarks, the relation and comparison with existing benchmarking may need more detailed descriptions, which could be put in the appendix.

**Additional Feedback:**

All points are covered in the main response.

**Documentation:**

This paper claims that the dataset and benchmarking can be downloaded from GitHub in the future.

**Relation To Prior Work:**

The authors claim that existing benchmarks are  (1) mostly constrained to plain CV, NLP, audio (2) time-consuming and environmentally unfriendly.

**Summary And Contributions:**

This paper introduces a comprehensive benchmarking for semi-supervised learning on vision, NLP and audio. And this paper provides an environmentally friendly and low-cost evaluation protocol with pre-training & fine-tuning paradigm, reducing the cost of SSL experiments. Additionally, this paper implements14 SSL algorithms and open-source a modular codebase and config files for easy reproduction of the reported results in this work.

---

> ### Author Response · Authors · 2022-08-17
> **Response to Reviewer ch9E**
>
> Thank you for the thoughtful comments. We now answer your questions.
>
> **Weaknesses1:** Although the experiment results are comprehensive, the insights and analysis are relatively limited. It would be better if the authors could provide more insights and discoveries from the experiment results. These insights are critical for inspiring SSL model designs in the future.
>
> **Answer:**  We have provided more insights and discoveries from the experiment results.
>
> **Weaknesses2:** It is suggested that what the benchmark can help for the SSL community could be described more clearly and explicitly in the paper.
>
> **Answer:**  We have revised our paper following your suggestions.
>
> **Weaknesses3:** Since there are already a lot of existing SSL benchmarks, the relation and comparison with existing benchmarking may need more detailed descriptions, which could be put in the appendix.
>
> **Answer:**
> We have included a related work section in the revised paper to show the difference between our work and existing ones.

---

### Official Review · Reviewer_cF9u · 2022-07-22
**Review: USB: A Unified Semi-supervised Learning Benchmark**

**Rating:** 6
**Confidence:** 4

**Strengths:**

In general, the paper is well motivated: No existing benchmark exists for semi/self-supervised learning algorithms which covers a wide range of domains, and approaches. The paper does a fairly good job at selecting efficient, yet relevant, classification tasks within the three target domains of language, vision, and audio. Given the performance on the USB benchmark, I would be relatively confident in the performance of the semi-supervised learning algorithm.

In addition, the paper has relatively strong analysis of the tested algorithms. I am much more confident in the performance and experiments given here than in many papers, and from the code provided, it is clear that the results are reproducible, and extensible. The framework for evaluation is also reasonable, and none of the methods for summarizing the performance over the tasks seem too opinionated.

I appreciate specifically that the paper includes multiple settings and seeds as part of the evaluation time and benchmarking process, which is often lacking in existing implementations.

I further really appreciate the fact that the amount of training time is taken into account, when exploring the

**Weaknesses:**

Generally, I'm worried about the scope of the paper, and the exact choices made for the benchmark. The USB benchmark as stated claims to be a "unified" benchmark for semi-supervised learning, however the chosen tasks largely ignore many of the tasks that individual communities consider challenging or difficult. More specifically, I'm concerned that the benchmark is limited to classification tasks, which are traditionally global in nature, and require little fine-grained detail to solve. Such a decision may lead to research prioritizing models which are designed for classification tasks, and ignore other equally important and interesting tasks.

- In vision, many applications of semi/self-supervised learning focus on fine-grained and zero-shot classification, which are mostly missing from the benchmark (Semi-Aves is the only included example which is even close). Additionally, semi/self supervised learning has been shown to be applicable in tasks in object detection, grounding, 3D vision, and image generation - which are all widely applicable, and interesting to explore.
- In NLP, the chosen datasets are more widely explored, but ignore important tasks such as language generation/general language modeling, summarization, and open-domain QA. Additionally, datasets like IMDB are largely considered solved in the supervised domain, which can make it hard to use as a differentiating task (even in SSL).
- In audio, the benchmarks focus on audio classification, but many other tasks are relevant including prosody classification, speaker recognition, automated speech recognition, and others (for a good summary of tasks, see the SUPERB benchmark, [46] in the paper).

**In particular, I would like to see a wider range of tasks in a "unified" benchmark, which closer reflect the current SOTA applications for semi and self-supervised learning.**

Another concern I have is with the chosen algorithms. As discussed in the limitations section of the paper, there are a significant number of SSL-based algorithms which are missing. Further, the chosen algorithms to be implemented are largely from a single direction (*-match) in the SSL space, which could bias future research towards algorithms structured in significant ways. **I would really like to see a wider range of field-representative algorithms, rather than a detailed exploration of single algorithm variants, included in the benchmark**

**Finally, there is no related work section in the paper, which makes it difficult to place in context.**

Some other minor comments:
- While Table 4 has a few interesting properties of each of the algorithms, there's no analysis of the impact of these properties on the USB benchmark scores. It would be interesting, and relevant, to explore how such properties impact the process.
- There is little discussion as to why each of the datasets was chosen beyond "they are in use in the field".
- I would really like to see some figures which demonstrate the correlation between the standard benchmark and the proposed USB benchmark. It's a bit hard to parse from the table if the results are correlated, and if they are, how closely they are. Related: I would appreciate some more discussion of the numerical correlations between the standard approach and the proposed approach, and I additionally wonder if a subset of the chosen experiments could be used to reduce training time, and would correlate as well or better with the full chosen experiment set. Is it possible to demonstrate that every benchmark chosen is necessary?
- The analysis of the TSNE in Figure-3 is somewhat superficial, and doesn't really add much to the paper. What makes for a "better" TSNE embedding is under-specified at best, and does not always indicate separability (see [here](https://distill.pub/2016/misread-tsne/)).
- The backbones that are chosen for the approach are somewhat arbitrary. It would be good to include a discussion of possible backbones, and to understand why each backbone was chosen.
- Performance on some of the benchmarks was a bit confusing, as it was presented as error-rate instead of accuracy (which is standard for many of the datasets shown).





**Additional Feedback:**

In summary, while I believe that this paper represents a solid technical effort and is globally well motivated, I am concerned about its utility as a benchmark, and novelty compared to existing work (which is hard to evaluate due to the distributed nature of the related work section). I would be more willing to accept a paper if it were re-scoped from "unified" to more correctly describe the target tasks, or if better motivation was included for the selection of each task, which was more aware of the trends in self/semi-supervised learning. I'm also somewhat concerned about the impact of this paper as a benchmark, as it might bias future research in ways that are not necessarily in the best interests of the field - such as an overwhelming focus on classification over other tasks, or a focus on exploration of *Match style algorithms over other directions in the field. That being said, I really appreciate the attempt to unify SSL benchmarks under one approach (which I believe is necessary, and important work), as well as the attention to the training time and technical correctness that this paper presents.

**Clarity:**

The paper is sufficiently clear. That being said, I believe (in accordance with the weaknesses of the paper) that some of the claims of being a "unified" benchmark should be toned down, since the benchmark is hardly unified across a wide range of applications. Appendix C. is very detailed, and shows a lot of useful information. The related work section is not present, which, although not *strictly necessary* would be helpful for readers in understanding and contextualizing information quickly.

**Correctness:**

I am confident that the included code is correct, and the benchmark is sound. The experimental design seems correct, and I particularly appreciate the inclusion of multiple seeds in the benchmarking procedure. I believe that the experiments could be reproduced, and the code is clean enough to be used in a potential benchmark.

**Documentation:**

No dataset documentation included (but may not be necessary). The documentation for the benchmark in the submitted code is sufficient, but could use a bit more explanation on how to implement your own models. Some sections are still TODO, and should be updated before release.

**Ethics:**

I have no ethics concerns for this paper - however I have not reviewed ethics concerns in the chosen benchmark datasets. It would be good for the authors to include a summary of [dataset cards](https://arxiv.org/abs/2204.01075) (another format is given [here](https://huggingface.co/docs/datasets/v1.12.0/dataset_card.html)) for each of the chosen tasks, which would surface any ethical concerns with the chosen tasks.

**Relation To Prior Work:**

There is no related work section, which makes the paper difficult to contextualize with existing benchmarking procedures. It is not entirely clear how this work differs from other attempts at benchmarking (if such attempts exist). If no such attempt to benchmark semi-supervised learning has ever been made, a small related work section discussion existing methods and their specific drawbacks (beyond those superficially mentioned in the introduction) would be appreciated. The definition of the "typical protocol" is not really well motivated, and we have to take it at face value that such a protocol is typical, and widely in use.

**Summary And Contributions:**

This paper introduces USB, a benchmark for semi-supervised learning which covers classification problems in several domains including vision (CIFAR-100, STL-10, EuroSAT, TissueMNIST, Semi-Aves), language (IMDB, AG-News, Amazon Review, Yahoo! Answer, and Yelp) and audio (ESC-50, GTZAN, UrtaSound8k, FSDnoisy18K). This benchmark aims to cover more domains than the standard approaches for benchmarking, as well as to reduce the amount of computation required (37 GPU days vs. 335 GPU days for FixMatch). The paper also introduces code for 14 existing SSL algorithms, which are evaluated according to the benchmark. Notably, the paper shows that under the USB benchmark, FlexMatch gains 5 places in ranking over the previous CRMatch approach in computer vision.

The paper's major contributions are:
- Introducing a collection of tasks for benchmarking SSL
- Demonstrating, with some analysis, the performance of some SSL algorithms on this task

---

> ### Author Response · Authors · 2022-08-17
> **Response to Reviewer cF9u - Part1**
>
> Thank you for the thoughtful comments. We now answer your questions.
>
> **Ethics:** I have no ethics concerns for this paper - however, I have not reviewed ethics concerns in the chosen benchmark datasets. It would be good for the authors to include a summary of dataset cards for each of the chosen tasks, which would surface any ethical concerns with the chosen tasks.
>
> **Answer:** We have added a summary of dataset cards in our codebase.
>
> **Documentation:** The documentation for the benchmark in the submitted code is sufficient, but could use a bit more explanation on how to implement your own models. Some sections are still TODO, and should be updated before release.
>
> **Answer:** This part has already been made to Colab tutorials in USB code and we have added more explanations on how to implement our own models in the documentation.
>
>
> **Weaknesses 1:** Generally, I'm worried about the scope of the paper, and the exact choices made for the benchmark. The USB benchmark as stated claims to be a "unified" benchmark for semi-supervised learning, however, the chosen tasks largely ignore many of the tasks that individual communities consider challenging or difficult. More specifically, I'm concerned that the benchmark is limited to classification tasks, which are traditionally global in nature, and require little fine-grained detail to solve. Such a decision may lead to research prioritizing models which are designed for classification tasks and ignore other equally important and interesting tasks.
>
> **Answer:** Thank you very much for the detailed feedback. We agree that a unified benchmark should contain a wider range of tasks that individual communities consider challenging or difficult. As we discussed in Limitation in the paper, we will continuously evolve the benchmark in the future by adding more tasks, and also extending classification tasks we start from through ourselves and community efforts. Additionally, unified means the unification of different domains in our paper and we make it clear in the revision. Designing a unified benchmark for all kinds of tasks across all domains is very challenging and certainly beyond the scope of this one single paper. But we are committed to continuing working under USB to extend it to imbalanced SSL, openset SSL, Seq2Seq SSL, etc. We also modified the title to "USB: A Unified Semi-supervised Learning Benchmark for Classification" and toned down the paper accordingly. Please have a check.

---

> > ### Comment · Reviewer_cF9u · 2022-08-17
> > **Thanks for the detailed response**
> >
> > Thanks for the detailed response, which clearly addresses most of the concerns I have with this work. The clarifications in the paper are really welcome, and I appreciate the updated appendix sections, and detailed additional experiments.
> >
> > I do still believe that the motivation for the chosen datasets could be better discussed: For example, why Yahoo! Answer was chosen over a dataset like SQUAD, but as the authors have promised to support additional datasets, this is a minor concern.
> >
> > Given the response, I also plan to increase my score, as the updated paper/analysis is significantly improved.

---

> > > ### Author Response · Authors · 2022-08-18
> > > **Thanks for the response.**
> > >
> > > Thanks for the prompt response!
> > >
> > > We have added more discussions in the revision. We mostly followed previous work in SSL in the NLP literature, and we reported all of the datasets used in the existing work in our paper. One more reason that we did not include SQUAD in our existing datasets is that it is a multi-choice task, which is slightly different from a classification task in the sense that the choices vary from test instance to instance, but are not constant class labels across instances. We agree that datasets such as SQUAD and other tasks in GLUE are interesting, and will add more types of tasks and data in our continued effort to extend our benchmark.

---

> > > ### Author Response · Authors · 2022-08-22
> > > **Backbone Results - Part1**
> > >
> > > We have now obtained error rate results of different pre-trained backbones, which are shown below. We will continuously update the results, and release our code, models, and logs at different checkpoints in USB.
> > >
> > > **CV**
> > >
> > > To compare different backbones on CV tasks, we fine-tune pre-trained public Swin-Transformer[1] with USB. We keep all hyper-parameters the same as in Table 16 and mainly evaluate on EuroSAT (32) and Semi-Aves (224). For EuroSAT, we change the input image size of pre-trained Swin-S from 224 to 32, and the window size from 7 to 4 to accommodate the adapted input image size. For Semi-Aves, we adopt the original Swin-S.
> > >
> > > From the results, one can observe, that on EuroSAT (32), as we adopt 224 pre-trained Swin-S and change its input and window size, the results are inferior to ViT-32 reported in the paper, whereas on Semi-Aves (224), the results are better than ViT-S. An interesting finding is that CoMatch performs relatively better with Swin-S while CrMatch performs worse. This also shows the importance of constantly updating the backbone as we promised.
> > >
> > >
> > > |                  | EuroSAT-20labels   | EuroSAT-40labels   | Semi-AVES  |
> > > |------------------|------------------------|---------------------|------------|
> > > |fullysupervised   | 1.86±0.10	        |1.86±0.10            | -          |
> > > |supervised        | 44.32±1.10	        |34.4±1.44          |38.76±0.21  |
> > > |pseudolabel       | 42.6±0.49	        |   32.79±1.54       | 38.50±0.73 |
> > > |meanteacher       | 35.85±1.95        |   19.62±3.28       |  33.37±0.06|
> > > |pimodel       | 	42.49±3.21        |    30.54±1.37      | 38.74±0.60|
> > > |vat       | 	  40.63±2.68      |    29.94±1.87      | 35.84±0.36 |
> > > |mixmatch       | 	 39.41±3.52       |    35.22±1.95      | 34.28±0.29 |
> > > |remixmatch       | 9.67±0.48	        |    7.45±0.63      | 27.8±0.32 |
> > > |adamatch       | 	 12.24±2.22       |      9.72±0.45    | 28.76±0.26 |
> > > |uda       | 	  18.15±5.70      |    12.09±1.26      | 29.28±0.20 |
> > > |fixmatch       | 	 17.19±3.46       |    12.57±1.28      |  28.88±0.22 |
> > > |flexmatch      | 	 10.46±1.20       |    9.06±1.80      | 30.19±0.51 |
> > > |dash       | 	  18.04±1.21      |     12.98±1.27     | 28.69±0.39 |
> > > |crmatch       | 	30.28±1.64        |    22.39±1.41      | 29.22±0.21 |
> > > |comatch     | 	   13.65±1.42     |     10.17±0.68     | 37.71±0.31 |
> > > |simmatch     | 	 11.19±1.01       |      10.65±1.64    | 28.55±0.13 |
> > >
> > > **NLP**
> > >
> > > For NLP tasks, we additionally experiment with RoBERTa[2]. We train RoBERTa using the same hyper-parameters reported in Table 17. RoBerta generally performs better than Bert as expected. The performance difference is both very close when using RoBerta or Bert.
> > > |                  | Yelp-250labels (RoBerta)   |  Yelp-1000labels (RoBerta)  |
> > > |------------------|--------------------|--------------------|
> > > |fullysupervised   | 29.15±0.12	        |29.15±0.12           |
> > > |supervised        | 42.56±1.15	        |	39.00±0.16          |
> > > |pseudolabel       | 48.26±0.02	        |   40.56±0.16       |
> > > |meanteacher       |49.41±0.03       |   44.36±1.04      |
> > > |pimodel       | 	49.16±2.04       |   42.93±0.88     |
> > > |vat       | 	 43.04±0.02      |    39.24±0.06     |
> > > |adamatch       | 	38.24±0.02     |    35.64±0.24    |
> > > |uda       | 	 40.13±0.15     |    	38.98±0.03    |
> > > |fixmatch       | 	 39.82±0.95       |   37.42±0.30     |
> > > |flexmatch      | 	39.11±0.02      |    36.84±0.01  |
> > > |dash       | 	 39.86±1.01     |     	36.23±0.21    |
> > > |crmatch       | 	40.08±1.28       |    35.85±0.38      |
> > > |comatch     | 	   39.95±0.86    |    	36.89±0.22    |
> > > |simmatch     | 	38.76±0.68      |    36.39±0.34   |

---

> > > ### Author Response · Authors · 2022-08-22
> > > **Backbone Results - Part2**
> > >
> > > **Audio**
> > >
> > > Due to the audio tasks setting in the current version of USB being built upon raw waveforms, there are not many pre-trained models available to use. We report the results of HuBert [3] and Wave2Vecv2.0 [4] for audio tasks to compare different backbones.
> > > The difference between these two backbones selected mainly lies in pre-training data. Wave2Vecv2.0 is pre-trained using raw human voice data and HuBert is an improved model with a discrete clustering target. Thus we can observe from the results, that on human voice tasks Superb-KS, Wave2Vecv2.0 has better performance, whereas, on other tasks, HuBert is more robust and outperforms Wave2Vecv2.0.
> > >
> > > |                  | keywordSpotting-50labels (HuBert)   | keywordSpotting-400labels (HuBert)  | FSDnoisy (Wave2Vec 2.0)  |
> > > |------------------|--------------------|--------------------|------------|
> > > |fullysupervised   | 2.41±0.15	        |2.41±0.15           | -          |
> > > |supervised        | 8.95±1.62	        |6.31±0.46          |33.54±1.65  |
> > > |pseudolabel       | 25.59±2.88	        |   13.02±2.47       | 35.23±0.78 |
> > > |meanteacher       |89.79±0.30       |   90.01±0.02      |  40.13±1.70 |
> > > |pimodel       | 	87.86±2.88        |   72.89±3.23     | 35.97±0.84 |
> > > |vat       | 	  2.27±0.07      |    2.43±0.02      | 34.21±0.31 |
> > > |adamatch       | 	8.17±4.24      |     2.36±0.07    | 50.83±29.48 |
> > > |uda       | 	  11.76±0.06     |    2.23±0.16    | 33.09±1.03 |
> > > |fixmatch       | 	 11.63±0.24       |    8.93±2.04      |  33.09±0.64 |
> > > |flexmatch      | 	 10.22±1.10       |    5.10±3.70   | 32.66±4.09 |
> > > |dash       | 	  11.88±0.15     |     	8.25±4.22     | 33.02±1.39 |
> > > |crmatch       | 	5.85±1.19       |    3.66±0.33      | 30.48±0.65 |
> > > |comatch     | 	   15.96±1.02    |     10.34±1.52     | 30.24±0.55 |
> > > |simmatch     | 	 9.43±0.63      |     5.47±2.72    | 29.57±0.52 |
> > >
> > >
> > > [1] Liu, Ze, et al. "Swin transformer: Hierarchical vision transformer using shifted windows." Proceedings of the IEEE/CVF International Conference on Computer Vision. 2021.
> > >
> > > [2] Liu, Yinhan, et al. "RoBERTa: A Robustly Optimized BERT Pretraining Approach." (2019).
> > >
> > > [3] Wei-Ning, Hsu, et al. "HuBERT: Self-Supervised Speech Representation Learning by Masked Prediction of Hidden Units." (2021)
> > >
> > > [4] Alexei, Baevski, et al. "wav2vec 2.0: A Framework for Self-Supervised Learning of Speech Representations." (2020)

---

> > > ### Author Response · Authors · 2022-08-23
> > > **Revision on backbone results**
> > >
> > > We have added the discussion part about different backbones in the Appendix of the revised paper. Please check the revised paper.

---

> > > ### Author Response · Authors · 2022-08-24
> > > **Followup On Revision and Responses**
> > >
> > > Dear reviewer cF9u，
> > >
> > > Thanks a lot for devoting much time to make USB better. We would like to send this friendly reminder and are very willing to discuss more deeply. If our response and revision address your concerns, could you please reconsider the score as you mentioned?

---

> ### Author Response · Authors · 2022-08-17
> **Response to Reviewer cF9u - Part2**
>
> **Weaknesses 2:** Another concern I have is with the chosen algorithms. As discussed in the limitations section of the paper, there are a significant number of SSL-based algorithms which are missing. Further, the chosen algorithms to be implemented are largely from a single direction (*-match) in the SSL space, which could bias future research towards algorithms structured in significant ways. I would really like to see a wider range of field-representative algorithms, rather than a detailed exploration of single algorithm variants, included in the benchmark.
>
>
> **Answer:** The core ideas behind different algorithms can be significantly different and they are definitely *not* variants of a certain algorithm. In fact, We think that researchers just named their algorithms by following FixMatch. For instance, ReMixMatch and MixMatch introduce MixUp into SSL and even do not include the thresholding mechanism which is the core property of FixMatch. USB covers all streams of algorithms listed by a recent survey [1] for classification except generative methods and GCN-based methods. This is because GCN-based methods are designed for graph/node classification tasks, which are radically different from image classification tasks and SSL algorithms except GCN-based methods are also not suitable for graph/node classification tasks. We did not include GAN-based methods because their performance on the existing benchmark is inferior to most consistency-based methods (R3-CGAN has a test error rate of 6.69% on CIFAR-10 with 4000 labels while FixMatch in the same year achieves 4.21%) and applying pre-trained ViT on them is not trivial and often leads to unstable training. That being said, we will consider making a graph/node classification benchmark for semi-supervised learning in the future.
>
> We provide results of MarginGAN, which is the most recent GAN-based method with code available, using the TorchSSL benchmark in the table below:
>
> |                  | CIFAR10-40 Labels        |  CIFAR10-250 Labels        | CIFAR10-4000 Labels       |   | CIFAR100-400 Labels        | CIFAR100-2500 Labels       | CIFAR100-10000 Labels     |
> |------------------|------------|------------|------------|---|------------|------------|------------|
> | FullySupervised  | 95.38±0.05 | 95.39±0.04 | 95.38±0.05 |   | 80.70±0.09 | 80.70±0.09 | 80.73±0.05 |
> | PiModel          | 25.66±1.76 | 53.76±1.29 | 86.87±0.59 |   | 13.04±0.80 | 41.20±0.66 | 63.35±0.00 |
> | PseudoLabel      | 25.39±0.26 | 53.51±2.20 | 84.92±0.19 |   | 12.55±0.85 | 42.26±0.28 | 63.45±0.24 |
> | PseudoLabel_Flex | 26.26±1.96 | 53.86±1.81 | 85.25±0.19 |   | 14.28±0.46 | 43.88±0.51 | 64.40±0.15 |
> | MeanTeacher      | 29.91±1.60 | 62.54±3.30 | 91.90±0.21 |   | 18.89±1.44 | 54.83±1.06 | 68.25±0.23 |
> | VAT              | 25.34±2.12 | 58.97±1.79 | 89.49±0.12 |   | 14.80±1.40 | 53.16±0.79 | 67.86±0.19 |
> | MixMatch         | 63.81±6.48 | 86.37±0.59 | 93.34±0.26 |   | 32.41±0.66 | 60.24±0.48 | 72.22±0.29 |
> | ReMixMatch       | 90.12±1.03 | 93.70±0.05 | 95.16±0.01 |   | 57.25±1.05 | 73.97±0.35 | 79.98±0.27 |
> | UDA              | 89.38±3.75 | 94.84±0.06 | 95.71±0.07 |   | 53.61±1.59 | 72.27±0.21 | 77.51±0.23 |
> | UDA_Flex         | 94.56±0.52 | 94.98±0.07 | 95.76±0.06 |   | 54.83±1.88 | 72.92±0.15 | 78.09±0.10 |
> | FixMatch         | 92.53±0.28 | 95.14±0.05 | 95.79±0.08 |   | 53.58±0.82 | 71.97±0.16 | 77.80±0.12 |
> | FlexMatch        | 95.03±0.06 | 95.02±0.09 | 95.81±0.01 |   | 60.06±1.62 | 73.51±0.20 | 78.10±0.15 |
> | MarginGAN        | 17.78±1.94 | 53.02±11.36| 92.87±0.12 |   | 13.70±1.80 | 48.01±0.46 | 69.54±0.31 |
>
>
> MarginGAN's performance at a low data regime (40 labels on CIFAR10 and 400 labels on CIFAR-100) is much lower than that of consistency-based methods.
>
> [1] Yang, Xiangli, et al. "A survey on deep semi-supervised learning." arXiv preprint arXiv:2103.00550 (2021).

---

> ### Author Response · Authors · 2022-08-17
> **Response to Reviewer cF9u - Part3**
>
> **Weaknesses 3:** Finally, there is no related work section in the paper, which makes it difficult to place in context.
>
> **Answer:** We have added a dedicated related work section in the revision.
>
> **Weaknesses 4:** While Table 4 has a few interesting properties of each of the algorithms, there's no analysis of the impact of these properties on the USB benchmark scores. It would be interesting, and relevant, to explore how such properties impact the process.
>
> **Answer:**  We will try to add more analysis. But the main contribution of our work is to arouse people's awareness of the importance of diverse domains in the SSL benchmark and make SSL research more environmentally friendly. The analysis of properties of algorithms will be an interesting future direction that we will definitely explore.
>
> **Weaknesses 5:** There is little discussion as to why each of the datasets was chosen beyond "they are in use in the field".
>
> **Answer:**  The datasets are chosen based on the following considerations: (1) the tasks should be diverse to cover multiple domains; (2) the tasks should be challenging, leaving room for improvement; (3) the training is reasonably environmentally friendly and affordable to research labs (particularly in academia); (4) we remove toy datasets where SSL algorithms present very close performance (usually differs in 0.0x). We agree that there are other important tasks and datasets which continually evolve. However, as we stated repeatedly, USB is not a static project, and we will expand our datasets over time for adapting to the rapid development of SSL.
>
> **Weaknesses 6:** I would really like to see some figures which demonstrate the correlation between the standard benchmark and the proposed USB benchmark. It's a bit hard to parse from the table if the results are correlated and if they are, how closely they are. Related: I would appreciate some more discussion of the numerical correlations between the standard approach and the proposed approach, and I additionally wonder if a subset of the chosen experiments could be used to reduce training time, and would correlate as well or better with the full chosen experiment set. Is it possible to demonstrate that every benchmark chosen is necessary?
>
> **Answer:**  Thank you for the question. We add a section in the Appendix to address your concern. Please check Appendix Section C: Correlation between TorchSSL and USB.
>
> **Weaknesses 7:** The analysis of the TSNE in Figure-3 is somewhat superficial, and doesn't really add much to the paper. What makes for a "better" TSNE embedding is under-specified at best, and does not always indicate separability.
>
> **Answer:** Thank you for pointing out the drawbacks of TSNE plots. We did pay attention to the potential pitfalls when generating the TSNE plots. The plots in the paper are generated using the same set of default parameters in SciKit. Therefore, they should still be able to serve as a comparison, to demonstrate that the pre-trained ViT has the most separable feature space after training in some cases. However, we fully agree with you that TSNE plots should not be over-interpreted and should never be used to replace other more grounded evaluation metrics. That's why we briefly mentioned this qualitative results after the discussion in Figure 1 and 2.

---

> ### Author Response · Authors · 2022-08-17
> **Response to Reviewer cF9u - Part4**
>
> **Weaknesses 8:** The backbones that are chosen for the approach are somewhat arbitrary. It would be good to include a discussion of possible backbones and to understand why each backbone was chosen.
>
> **Answer:**  We are running experiments to provide more experiments with different backbones. Actually, currently, we select the SOTA and well-known backbones for each domain. For CV, modified ViT is pre-trained and used as we find simply using pre-trained ViT of 224 image size and a patch size of 16 would lead to strong overfitting on USB CV tasks. Except for ViT, we also tried pre-trained WRN networks. On Audio tasks, we actually trained two backbones - HuBert and Wave2Vec, and select the best for each dataset. We indeed just used BERT-Base for NLP tasks.
> For each domain (CV, NLP, and Audio), we are trying to do more experiments to provide the evaluation of one more pre-trained backbone.
>
> **Weaknesses 9:** Performance on some of the benchmarks was a bit confusing, as it was presented as error-rate instead of accuracy (which is standard for many of the datasets shown).
>
> **Answer:**  The tradition of SSL papers (ReMixMatch, UDA, FixMatch, FlexMatch) presents error rate as default. We have revised our paper to make it clear.
>
> **Clarity 1:** The paper is sufficiently clear. That being said, I believe (in accordance with the weaknesses of the paper) that some of the claims of being a "unified" benchmark should be toned down since the benchmark is hardly unified across a wide range of applications.
>
> **Answer:**  Thanks for your suggestions. Unified means the unification of domains in our paper. We have made it clear in the revision. We are not sure whether this change removes your concerns about the claim. Note that we did not change the term unified after thinking of alternatives such as general, diverse and multi-domain. If you still have concerns about toning down the name, let us know!
>
> **Clarity 2:** The related work section is not present, which, although not strictly necessary would be helpful for readers in understanding and contextualizing information quickly.
>
> **Answer:**  We have added a related work section in the revision.
>
> **Relation To Prior Work:** Related work section
>
> **Answer:** We have added a dedicated related work section to make it clear.
>
>
> **Additional Feedback:**
> In summary, while I believe that this paper represents a solid technical effort and is globally well motivated, I am concerned about its utility as a benchmark, and novelty compared to existing work (which is hard to evaluate due to the distributed nature of the related work section). I would be more willing to accept a paper if it were re-scoped from "unified" to more correctly describe the target tasks, or if the better motivation was included for the selection of each task, which was more aware of the trends in self/semi-supervised learning. I'm also somewhat concerned about the impact of this paper as a benchmark, as it might bias future research in ways that are not necessarily in the best interests of the field - such as an overwhelming focus on classification over other tasks, or a focus on exploration of *Match style algorithms over other directions in the field. That being said, I really appreciate the attempt to unify SSL benchmarks under one approach (which I believe is necessary, and important work), as well as the attention to the training time and technical correctness that this paper presents.
>
> **Answer:**  Thanks for this comment. USB will be a dynamic benchmark. We have already done some experiments on smei-supervised regression tasks. The dataset we chose is The Boston Housing Dataset[1]. We sample 10% from the original dataset to form the labeled dataset, 60% to form the unlabeled dataset, and 30% to form the test dataset. Note that we only did experiments using Π Model and Mean Teacher since other SSL algorithms rely on the probability generated by the SoftMax layer which regression tasks do not have. The results are as follows:
>
> |    Method            |   Mean Absolute Error         |
> |----------------------|-------------------------------|
> |   Π Model            |       4.33                    |
> | Mean Teacher         |       4.49                    |
>
> [1] https://www.cs.toronto.edu/~delve/data/boston/bostonDetail.html

---

### Official Review · Reviewer_tMH5 · 2022-07-27
**an interesting paper, but need more insights/illustrations**

**Rating:** 6
**Confidence:** 3
**Correctness:** The evaluation methods and experiment…

**Strengths:**

The problem is interesting and important. The SSL problem has been widely studied in the machine learning community. It is a fundamental problem in CV and NLP.

The authors implement 14 well-known SSL algorithms and put them into an open-sourced codebase for easy comparison. Comprehensive experiments clearly demonstrate the differences.

The authors provide analysis from the experiments such as the difference between training from scratch and using pre-training in SSL, and the effectiveness of SSL when using the state-of-the-art neural models as the backbones.


**Weaknesses:**

The authors should be more clear about the motivation, and why this work/open-sourced codebase is useful for the community. For example, in the introduction, the authors mentioned that existing benchmarks are mostly constrained to CV tasks, but do not clearly illustrate why this is a problem, and what advantages can be gained by building a unified SSL benchmark.

Lack of insights or explanations for the experimental results. For example, in section 4.3, when answering why should we evaluate an SSL algorithm on diverse tasks across domains. The authors said, “the differences between ranks of SSL algorithms in different domains cannot be ignored, hence it is crucial to introduce diverse tasks from multiple domains when evaluating an SSL algorithm”. Why can it not be ignored? After introducing diverse tasks, what insights can we get?


**Additional Feedback:**

Could you summarize the insights of this paper? What can we learn by comparing so many SSL algorithms across different tasks?

**Clarity:**

The tables and figures are clear and supportive of the experiments. However, the authors need to provide more illustrations and clarifications about the motivation.

**Documentation:**

Good.

**Ethics:**

N/A.

**Relation To Prior Work:**

The authors discussed the difference compared to previous contributions.

**Summary And Contributions:**

The authors propose USB: a Unified SSL Benchmark to facilitate general SSL research. USB offers benchmarks across five CV datasets, five NLP datasets, and five Audio datasets. By using pre-trained Vision Transformers, the authors found that it can largely decrease the training time when evaluating the performance of an SSL algorithm. The authors implement 14 SSL algorithms and will open-source the codebase.

---

> ### Author Response · Authors · 2022-08-17
> **Response to Reviewer tMH5**
>
> Thank you for the thoughtful comments. We now answer your questions.
>
> **Weakness 1:** The authors should be more clear about the motivation, and why this work/open-sourced codebase is useful for the community. For example, in the introduction, the authors mentioned that existing benchmarks are mostly constrained to CV tasks, but do not clearly illustrate why this is a problem, and what advantages can be gained by building a unified SSL benchmark.
>
> **Answer:**  We fully agree and will try our best to improve our writing to make our motivation clear. There are mainly two reasons why we focus beyond the CV area:
> - First, from the *application* perspective, researchers in both NLP and Audio communities are getting more and more interested in SSL [1,2,3,4]. However, only a few algorithms such as UDA have been discussed in NLP, and it remains uncertain whether other algorithms such as FlexMatch and SimMatch work also. In a nutshell, there still lacks discussions about whether SSL methods that work well on CV tasks still work on NLP and Audio tasks. To answer the above question, we aim to build a unified SSL benchmark to provide intuitive cognition on all SSL algorithms across different domains.
> - Second, from the *SSL algorithm research* perspective, different domains of datasets certainly pose different challenges to the algorithms, thus, designing a *diverse* testbed will help us develop more robust algorithms and help researchers better analyze their approaches to different datasets. While CV datasets are widely used in most ML algorithms, we think adding NLP and Audio datasets will enlarge that diversity and become a better testbed.
>
> **Weakness 2:** Lack of insights or explanations for the experimental results. For example, in section 4.3, when answering why should we evaluate an SSL algorithm on diverse tasks across domains. The authors said, “the differences between ranks of SSL algorithms in different domains cannot be ignored, hence it is crucial to introduce diverse tasks from multiple domains when evaluating an SSL algorithm”. Why can it not be ignored? After introducing diverse tasks, what insights can we get?
>
> **Answer:**  We have expanded the discussion in our revision. To our humble knowledge, we are the first to discuss whether current SSL methods that work well on CV tasks still work on NLP and Audio tasks. The insights behind different SSL methods across different domains should be an important future direction.
>
> **Clarity:** The tables and figures are clear and supportive of the experiments. However, the authors need to provide more illustrations and clarifications about the motivation.
>
> **Answer:**  Please refer to the answer for Weakness 1.
>
> [1] Chen, Jiaao, Zichao Yang, and Diyi Yang. "MixText: Linguistically-Informed Interpolation of Hidden Space for Semi-Supervised Text Classification." Proceedings of the 58th Annual Meeting of the Association for Computational Linguistics. 2020.
>
> [2] Li, Changchun, Ximing Li, and Jihong Ouyang. "Semi-Supervised Text Classification with Balanced Deep Representation Distributions." Proceedings of the 59th Annual Meeting of the Association for Computational Linguistics and the 11th International Joint Conference on Natural Language Processing (Volume 1: Long Papers). 2021.
>
> [3] Lu, Kangkang, et al. "Semi-Supervised Audio Classification with Consistency-Based Regularization." INTERSPEECH. 2019.
>
> [4] He, Gewen, et al. "Image2audio: Facilitating semi-supervised audio emotion recognition with facial expression image." Proceedings of the IEEE/CVF Conference on Computer Vision and Pattern Recognition Workshops. 2020.

---

### Official Review · Reviewer_NkKF · 2022-07-27
**More justification is needed for the experimental choices in the USB benchmark.**

**Rating:** 7
**Confidence:** 4

**Strengths:**

- The benchmark includes a large and diverse evaluation suite: 15 different tasks in 3 different domains, including the audio domain, for which USB is the first to evaluate SSL on.
- 14 different SSL algorithms are implemented and compared on the benchmarks (save for those algorithms which do not make sense to implement in certain modalities).
- The benchmark is constructed with computational expense in mind, such that it requires much fewer GPU-hours to evaluate a method across the suite of tasks compared to previous SSL benchmarks.
- Using pretrained backbones is an important axis of variation missing in previous benchmarks, and USB investigates the impact of the use of a pretrained backbones, finding that most SSL algorithms become more efficient.
- USB compares the rankings of different algorithms across the different domains. I believe this analysis/result is the first for SSL algorithms to be compared between domains.

**Weaknesses:**

My main criticism of this work is that there is little justification given for a couple core experimental choices in the USB benchmark. This is a considerable missed opportunity, since USB is positioned as an compute-friendly SSL benchmark that academics or other researchers can hill-climb on. As such, this work can do a better job of convincing the reader that results from this benchmark will transfer to other settings, wherever possible, by performing additional experimental validation. Let me be concrete:

- The choice of pretraining backbone. I do agree that it is reasonable to leverage pretrained backbones as a way to improve the performance of SSL. However, this is a significant departure from the standard SSL setting and deserves some justification. Why are the specific pretrained ViT, BERT, or Wav2Vec defined in USB chosen (is there a reason they are better than alternatives)? Do the rankings or conclusions of which SSL algorithm is best change if the pretrained backbone is changed (for a different architecture, or one trained on a different dataset)? If the choice of pretraining backbone has a large influence on the SSL algorithm ranking, this should be discussed in more detail. If not, then the experiments need to be presented to inspire confidence in the current choice of backbone.

- Why is ImageNet excluded from USB? This seems like an odd choice, especially since *most* SSL work has been developed for ImageNet, and we know ImageNet is a reasonable dataset to hill-climb on since many previous works have shown that ImageNet gains routinely transfer to other datasets. There is some discussion about computational cost for the reason, but its exclusion is odd given that USB positions itself as "Unified." One way around this would be to conduct an evaluation on ImageNet and compare the rankings to the CV portion of USB, and if the rankings are consistent, this would be evidence that ImageNet results closely track the rest of USB-CV and thus perhaps redundant/not needed. But this needs to be run & shown.

Another weakness of this work is the choice of overall metric. Having a rank as the overall metric seems to be a very poor choice since the score for any method depends on the rest of the methods implemented in the benchmark at that point in time. If two new methods A and B were proposed at the same time separately, there would be no way to compare which of A and B is better by looking at their reported ranks until they were both implemented into the USB codebase and the ranks were recalculated. This seems like a major downside. What about just using average accuracy or something similar? It's a much more standard reporting metric and allows time-independent comparisons.

**Additional Feedback:**

I would be happy to increase my score, if the changes I outlined are made and depending on the result of the discussion.

***************** UPDATE AFTER REBUTTAL *****************************

I had 3 main concerns, each of which was satisfactorily answered by the authors:

1. Choice of pretraining backbone. The authors ran ablations for different backbones for each of the CV/NLP/Audio domains. It seems like relative algorithm ordering can indeed vary a bit in the top performing algorithms (ranks 1-6). While this is unideal for a benchmark like USB, the authors are willing to include a discussion of these ablations in the text, as well as have promised to update the benchmark regularly to new backbones. Together, this response seems satisfactory to me. While there may be a larger discussion around whether algorithm design or backbone improvements drive the lion's share of improvements in SSL, I feel this is out of scope for the benchmark as it is.

2. ImageNet results. The authors have run a few algorithms on ImageNet to compare with the rest of the CV USB datasets, and have included a compare/contrast discussion in the main text. This seems reasonable to me.

3. Better metrics. The authors have included mean error rate as a metric in addition to rank. I think this addition significantly improves results comparison for the paper.

Overall, all my concerns have been satisfactorily answered, and I believe this benchmark will be a solid contribution to the community, especially to those on low resource budgets.

**Clarity:**

The clarity of the writing and presentation of results could be significantly improved in parts. To be concrete:

- In the abstract, the last line is very unclear. Is it trying to highlight the evaluation speedup? Is it trying to highlight the dataset diversity? Two different pieces are being mentioned and it's hard to understand what contributes to the speedup if the two benchmarks are different and gpu-days are incomparable.

- There is a lot of space devoted to talking about the existing protocol. As a reader, having lots of information that isn't central to the story of USB was distracting. I think Tables 3 and 5 are useful but can be safely moved to the appendix.

- The last paragraph on page 4. This was hard to get through. I'm not sure what purpose this paragraph serves - it's just Table 4 written in prose. I would suggest using this space to talk about the different columns of Table 4 and what they mean instead.

**Correctness:**

For hyperparameter tuning, the appendix seems to imply that the parameters of FixMatch were tuned for each dataset, and then those parameters were used for evaluation of the rest of the SSL algorithms. Is this correct? If so, this would seem to greatly bias the FixMatch results - why aren't hyperparameters tuned for each method individually?

**Documentation:**

There was no maintenance plan specified. I would be curious to know how the authors plan to support the codebase in the future, who will provide support, where the resources for evaluations of new algorithms will come from, etc.

**Ethics:**

There are no ethics issues.

**Relation To Prior Work:**

I would suggest mentioning TorchSSL whenever the text says "existing popular protocol"

**Summary And Contributions:**

This work proposes a new benchmark for semi-supervised learning, USB, which covers evaluation on 15 different tasks in 3 different domains: vision, NLP, and audio. The benchmark is constructed to not be too computationally expensive to run. Additionally, a number of different SSL algorithms are compared on the benchmark and there is some analysis performed about which methods succeed.

---

> ### Author Response · Authors · 2022-08-17
> **Response to Reviewer NkKF - Part1**
>
> Thank you for the thoughtful comments. We now answer your questions.
>
> **Documentation:** There was no maintenance plan specified. I would be curious to know how the authors plan to support the codebase in the future, who will provide support, where the resources for evaluations of new algorithms will come from, etc.
>
> **Answer:**  We have added this in the revision. Microsoft Research Asia (MSRA) will provide both the support and resources for evaluations of new algorithms. Additionally, our team has been working closely via remote services during the development of USB and we certainly will continue expanding the collaboration to the whole SSL community just like other popular open-source communities. It takes effort to build such a strong team to consistently support this project, but we will try our best and look forward to making USB better continuously.
>
> **Weakness 1:** The choice of pretraining backbone. I do agree that it is reasonable to leverage pre-trained backbones as a way to improve the performance of SSL. However, this is a significant departure from the standard SSL setting and deserves some justification. Why are the specific pre-trained ViT, BERT, or Wav2Vec defined in USB chosen (is there a reason they are better than alternatives)? Do the rankings or conclusions of which SSL algorithm is the best change if the pre-trained backbone is changed (for a different architecture, or one trained on a different dataset)? If the choice of pretraining backbone has a large influence on the SSL algorithm ranking, this should be discussed in more detail. If not, then the experiments need to be presented to inspire confidence in the current choice of backbone.
>
> **Answer:**
> For CV, modified ViT is pre-trained and used as we find simply using ViT of 224 image size would lead to strong overfitting on USB CV tasks. Except for ViT, we also tried pre-trained WRN networks. On Audio tasks, we actually trained two backbones - HuBert and Wave2Vec, and select the best for each dataset. We indeed just used BERT-Base for NLP tasks.
> For each domain (CV, NLP, and Audio), we are trying to do more experiments to provide the evaluation of more pre-trained backbones. Hope we can get the results before the rebuttal ends. In addition, as mentioned in the submission, USB is intended to be dynamically evolving, where more datasets and results will be added through our and community efforts also.
>
> **Weakness 2:** Why is ImageNet excluded from USB? This seems like an odd choice, especially since *most* SSL work has been developed for ImageNet, and we know ImageNet is a reasonable dataset to hill-climb on since many previous works have shown that ImageNet gains routinely transfer to other datasets. There is some discussion about the computational cost for the reason, but its exclusion is odd given that USB positions itself as "Unified." One way around this would be to conduct an evaluation on ImageNet and compare the rankings to the CV portion of USB, and if the rankings are consistent, this would be evidence that ImageNet results closely track the rest of USB-CV and thus perhaps redundant/not needed. But this needs to be run & shown.
>
> **Answer:** The reason that we exclude ImageNet from USB is that ImageNet is large (>100GB) and it might not be easy for small academic labs or personal researchers to run SSL experiments with reasonable cost. However, we fully agree with your point that ImageNet is a reasonable dataset for hill-climbing. To verify the ranking is consistent on ImageNet, we have conducted ImageNet experiments of UDA, FixMatch, FlexMatch, CoMatch, and SimMatch with 1w and 10w labels (10 and 100 labels per class). Each experiment requires 5 days of training on 16 A100 GPUs. We use MAE pre-trained ViT-B as the backbone for ImageNet experiments. You can find more details in the Appendix of the latest revision.
>
> The evaluation of Top-1 accuracy on ImageNet and the rankings to the CV portion of USB are shown in the table below. The rankings of SSL algorithms are consistent, which implies that ImageNet results are closely reproduced by USB-CV and thus perhaps redundant as you conjectured.
>
> | Method 	| 1w Labels 	| 10w Labels 	| Rank 	|
> |---|---|---|---|
> | UDA 	| 38.62 	| 62.37 	| 5 	|
> | FixMatch 	| 37.93 	| 62.88 	| 4 	|
> | FlexMatch 	| 39.13 	| 63.09 	| 3 	|
> | CoMatch 	| 44.32 	| 65.80 	| 2 	|
> | SimMatch 	| 46.48 	| 67.61 	| 1 	|

---

> > ### Comment · Reviewer_NkKF · 2022-08-17
> > **Appreciate the follow-up experiments and modifications**
> >
> > Thank you for the thorough response to my questions, as well as for the other reviewers as well. It seems clear that great care and attention has been put into the experimental results of this benchmark.
> >
> > The ImageNet results are very interesting. UDA and Fixmatch are near the bottom, similar to USB. It also seems that SimMatch is still marked as one of the top, but it's surprising that CoMatch does so well on ImageNet (almost as good as SimMatch, looking at the accuracies) when it ranked only 9th on the USB benchmark. Also, while FlexMatch is the best on USB, it's pretty firmly behind CoMatch and SimMatch on ImageNet. So, I wouldn't really say the ImageNet results here are redundant with USB (unless I'm interpreting the results wrong). There are potentially 2 paths forward here: adding these results to the text with a discussion of the discrepancies, or adding ImageNet as one of the CV benchmarks to USB. I look forward to hearing what the authors think.
> >
> > I would also be very curious to know the results for the effect of different pretrained backbones. I also thank the authors for various other modifications - adding mean error rate, improving clarity and discussion, etc.
> >
> > I plan to significantly increase my score multiple points given these edits. Please do send a followup when the backbones results are available (or whatever portion of it is available before rebuttal period ends).

---

> > > ### Author Response · Authors · 2022-08-22
> > > **Backbone Results - Part1**
> > >
> > > We have now obtained error rate results of different pre-trained backbones, which are shown below. We will continuously update the results, and release our code, models, and logs at different checkpoints in USB.
> > >
> > > **CV**
> > >
> > > To compare different backbones on CV tasks, we fine-tune pre-trained public Swin-Transformer[1] with USB. We keep all hyper-parameters the same as in Table 16 and mainly evaluate on EuroSAT (32) and Semi-Aves (224). For EuroSAT, we change the input image size of pre-trained Swin-S from 224 to 32, and the window size from 7 to 4 to accommodate the adapted input image size. For Semi-Aves, we adopt the original Swin-S.
> > >
> > > From the results, one can observe, that on EuroSAT (32), as we adopt 224 pre-trained Swin-S and change its input and window size, the results are inferior to ViT-32 reported in the paper, whereas on Semi-Aves (224), the results are better than ViT-S. An interesting finding is that CoMatch performs relatively better with Swin-S while CrMatch performs worse. This also shows the importance of constantly updating the backbone as we promised.
> > >
> > >
> > > |                  | EuroSAT-20labels   | EuroSAT-40labels   | Semi-AVES  |
> > > |------------------|------------------------|---------------------|------------|
> > > |fullysupervised   | 1.86±0.10	        |1.86±0.10            | -          |
> > > |supervised        | 44.32±1.10	        |34.4±1.44          |38.76±0.21  |
> > > |pseudolabel       | 42.6±0.49	        |   32.79±1.54       | 38.50±0.73 |
> > > |meanteacher       | 35.85±1.95        |   19.62±3.28       |  33.37±0.06|
> > > |pimodel       | 	42.49±3.21        |    30.54±1.37      | 38.74±0.60|
> > > |vat       | 	  40.63±2.68      |    29.94±1.87      | 35.84±0.36 |
> > > |mixmatch       | 	 39.41±3.52       |    35.22±1.95      | 34.28±0.29 |
> > > |remixmatch       | 9.67±0.48	        |    7.45±0.63      | 27.8±0.32 |
> > > |adamatch       | 	 12.24±2.22       |      9.72±0.45    | 28.76±0.26 |
> > > |uda       | 	  18.15±5.70      |    12.09±1.26      | 29.28±0.20 |
> > > |fixmatch       | 	 17.19±3.46       |    12.57±1.28      |  28.88±0.22 |
> > > |flexmatch      | 	 10.46±1.20       |    9.06±1.80      | 30.19±0.51 |
> > > |dash       | 	  18.04±1.21      |     12.98±1.27     | 28.69±0.39 |
> > > |crmatch       | 	30.28±1.64        |    22.39±1.41      | 29.22±0.21 |
> > > |comatch     | 	   13.65±1.42     |     10.17±0.68     | 37.71±0.31 |
> > > |simmatch     | 	 11.19±1.01       |      10.65±1.64    | 28.55±0.13 |
> > >
> > > **NLP**
> > >
> > > For NLP tasks, we additionally experiment with RoBERTa[2]. We train RoBERTa using the same hyper-parameters reported in Table 17. RoBerta generally performs better than Bert as expected. The performance difference is both very close when using RoBerta or Bert.
> > > |                  | Yelp-250labels (RoBerta)   |  Yelp-1000labels (RoBerta)  |
> > > |------------------|--------------------|--------------------|
> > > |fullysupervised   | 29.15±0.12	        |29.15±0.12           |
> > > |supervised        | 42.56±1.15	        |	39.00±0.16          |
> > > |pseudolabel       | 48.26±0.02	        |   40.56±0.16       |
> > > |meanteacher       |49.41±0.03       |   44.36±1.04      |
> > > |pimodel       | 	49.16±2.04       |   42.93±0.88     |
> > > |vat       | 	 43.04±0.02      |    39.24±0.06     |
> > > |adamatch       | 	38.24±0.02     |    35.64±0.24    |
> > > |uda       | 	 40.13±0.15     |    	38.98±0.03    |
> > > |fixmatch       | 	 39.82±0.95       |   37.42±0.30     |
> > > |flexmatch      | 	39.11±0.02      |    36.84±0.01  |
> > > |dash       | 	 39.86±1.01     |     	36.23±0.21    |
> > > |crmatch       | 	40.08±1.28       |    35.85±0.38      |
> > > |comatch     | 	   39.95±0.86    |    	36.89±0.22    |
> > > |simmatch     | 	38.76±0.68      |    36.39±0.34   |

---

> > > ### Author Response · Authors · 2022-08-22
> > > **Backbone Results - Part2**
> > >
> > > **Audio**
> > >
> > > Due to the audio tasks setting in the current version of USB being built upon raw waveforms, there are not many pre-trained models available to use. We report the results of HuBert [3] and Wave2Vecv2.0 [4] for audio tasks to compare different backbones.
> > > The difference between these two backbones selected mainly lies in pre-training data. Wave2Vecv2.0 is pre-trained using raw human voice data and HuBert is an improved model with a discrete clustering target. Thus we can observe from the results, that on human voice tasks Superb-KS, Wave2Vecv2.0 has better performance, whereas, on other tasks, HuBert is more robust and outperforms Wave2Vecv2.0.
> > >
> > > |                  | keywordSpotting-50labels (HuBert)   | keywordSpotting-400labels (HuBert)  | FSDnoisy (Wave2Vec 2.0)  |
> > > |------------------|--------------------|--------------------|------------|
> > > |fullysupervised   | 2.41±0.15	        |2.41±0.15           | -          |
> > > |supervised        | 8.95±1.62	        |6.31±0.46          |33.54±1.65  |
> > > |pseudolabel       | 25.59±2.88	        |   13.02±2.47       | 35.23±0.78 |
> > > |meanteacher       |89.79±0.30       |   90.01±0.02      |  40.13±1.70 |
> > > |pimodel       | 	87.86±2.88        |   72.89±3.23     | 35.97±0.84 |
> > > |vat       | 	  2.27±0.07      |    2.43±0.02      | 34.21±0.31 |
> > > |adamatch       | 	8.17±4.24      |     2.36±0.07    | 50.83±29.48 |
> > > |uda       | 	  11.76±0.06     |    2.23±0.16    | 33.09±1.03 |
> > > |fixmatch       | 	 11.63±0.24       |    8.93±2.04      |  33.09±0.64 |
> > > |flexmatch      | 	 10.22±1.10       |    5.10±3.70   | 32.66±4.09 |
> > > |dash       | 	  11.88±0.15     |     	8.25±4.22     | 33.02±1.39 |
> > > |crmatch       | 	5.85±1.19       |    3.66±0.33      | 30.48±0.65 |
> > > |comatch     | 	   15.96±1.02    |     10.34±1.52     | 30.24±0.55 |
> > > |simmatch     | 	 9.43±0.63      |     5.47±2.72    | 29.57±0.52 |
> > >
> > >
> > > [1] Liu, Ze, et al. "Swin transformer: Hierarchical vision transformer using shifted windows." Proceedings of the IEEE/CVF International Conference on Computer Vision. 2021.
> > >
> > > [2] Liu, Yinhan, et al. "RoBERTa: A Robustly Optimized BERT Pretraining Approach." (2019).
> > >
> > > [3] Wei-Ning, Hsu, et al. "HuBERT: Self-Supervised Speech Representation Learning by Masked Prediction of Hidden Units." (2021)
> > >
> > > [4] Alexei, Baevski, et al. "wav2vec 2.0: A Framework for Self-Supervised Learning of Speech Representations." (2020)

---

> > > > ### Comment · Reviewer_NkKF · 2022-08-22
> > > > **Response to backbone results**
> > > >
> > > > Thanks for including the backbone results. Looking over each of the charts for CV/NLP/Audio, it seems that across the tasks there is a pretty clear distinction between algorithms in the first half of the list (ranks 7-12) and the second half of the list (ranks 1-6). While switching out backbones doesn't seem to change membership of these two halves, it does seem like the relative orderings within the top half (ranks 1-6) can indeed vary a bit. This seems like a useful takeaway to know, and I thank the authors for running these ablations. While the ranking dependence on the backbone is unideal for a benchmark like USB, the authors have promised to keep updating the benchmark regularly, which seems satisfactory to me. I think the addition of mean error rate as a metric will also help alleviate this issue. In addition, the authors were willing to engage and run additional experiments in the first place, and this discussion will be useful for others to know down the line (the appendix hasn't been updated with these results yet but I assume that will happen for the camera ready). In light of this discussion, I have updated my review & score.

---

> > > > > ### Author Response · Authors · 2022-08-23
> > > > > **Revision on backbone results**
> > > > >
> > > > > Thanks for your appreciation of our work! We have added the discussion part about different backbones in the Appendix of the revised paper.

---

> > > ### Author Response · Authors · 2022-08-22
> > > **Response to Questions on ImageNet**
> > >
> > > Thanks a lot for the useful suggestions! We agree with you and have modified the discussion part in Appendix D. Besides, we have added the following sentence in the revised paper. "Besides, we highly recommend reporting ImageNet results since it is a reasonable dataset for hill-climbing. We also report and discuss ImageNet results in Appendix D."

---

> ### Author Response · Authors · 2022-08-17
> **Response to Reviewer NkKF - Part2**
>
> **Weakness 3:** Another weakness of this work is the choice of overall metric. Having a rank as the overall metric seems to be a very poor choice since the score for any method depends on the rest of the methods implemented in the benchmark at that point in time. If two new methods A and B were proposed at the same time separately, there would be no way to compare which of A and B is better by looking at their reported ranks until they were both implemented into the USB codebase and the ranks were recalculated. This seems like a major downside. What about just using average accuracy or something similar? It's a much more standard reporting metric and allows time-independent comparisons.
>
> **Answer:**  We have added the average error rate of each SSL algorithm in the revision of our paper. But the average error rate will be heavily affected by the number of tasks easy for each domain, and the rank of algorithms is additionally provided to demonstrate the comprehensive performance among different tasks. Besides, we use bold font to highlight the best error rate in order to show the performance on the specific dataset.
>
> **Correctness:** For hyperparameter tuning, the appendix seems to imply that the parameters of FixMatch were tuned for each dataset, and then those parameters were used for the evaluation of the rest of the SSL algorithms. Is this correct? If so, this would seem to greatly bias the FixMatch results - why aren't hyperparameters tuned for each method individually?
>
> **Answer:**  A short answer is that tuning hyperparameters for each algorithm on each dataset is computationally heavy and requires lots of computing resources. We select FixMatch to tune the base parameters because UDA, FlexMatch, CoMatch, SimMatch, and all algorithms developed after FixMatch all use the default hyperparameter of FixMatch. Besides, the parameter we tuned mainly is learning rate and weight decay, which do not concern algorithmic parameters. In addition, as USB aims to dynamically evolve through community effort, we foresee that more results will be reported to the popular methods.
>
>
> **Clarity 1:** In the abstract, the last line is very unclear. Is it trying to highlight the evaluation speedup? Is it trying to highlight the dataset diversity? Two different pieces are being mentioned and it's hard to understand what contributes to the speedup if the two benchmarks are different and GPU-days are incomparable.
>
> **Answer:** Yes, it highlights the evaluation speedup as well as the dataset diversity. Evaluation on USB requires fewer GPU days despite the diversified datasets. The speedup comes from the use of pre-trained models.
>
> **Clarity 2:** There is a lot of space devoted to talking about the existing protocol. As a reader, having lots of information that isn't central to the story of USB was distracting. I think Tables 3 and 5 are useful but can be safely moved to the appendix.
>
> **Answer:** Thanks for the suggestion. We have modified the paper accordingly. Note that the classic setting results are also from the USB codebase. We add a classic setting in the USB codebase to verify the correctness of implementation and for comparing USB tasks. The main contribution of this work is not to find the best algorithm of all SSL algorithms. As we repeatedly stated in the paper, our contribution is to reduce the cost of SSL and add NLP and Audio tasks to SSL benchmarks.
>
> **Clarity 3:** The last paragraph on page 4. This was hard to get through. I'm not sure what purpose this paragraph serves - it's just Table 4 written in prose. I would suggest using this space to talk about the different columns of Table 4 and what they mean instead.
>
> **Answer:** Thanks for the suggestion, we have added discussion concerning columns of Table 4 in the revised paper.
>
> **Relation To Prior Work:** I would suggest mentioning TorchSSL whenever the text says "existing popular protocol"
>
> **Answer:** Thanks for the suggestion, we have modified the paper accordingly. Please check the revised paper. But please note that TorchSSL does not include SimMatch, AdaMatch, CrMatch, Dash, and CoMatch. We report the results with our own codes and add more algorithms to the existing CV tasks (CIFAR, SVHN, STL10).

---

### Official Review · Reviewer_ffrp · 2022-07-27
**Review of USB: A Unified Semi-supervised Learning Benchmark**

**Rating:** 7
**Confidence:** 4
**Clarity:** This paper clearly describes the expe…

**Strengths:**

This paper is well motivated. As the authors point out, previous work typically trains deep neural networks from scratch, which is time-consuming and environmentally unfriendly. I applaud the author's efforts in experiments. To clearly and comprehensively clarify the strengths and weaknesses of different algorithms in different domains, they conduct quite enormous experiments. The authors also use nearly every dataset commonly used in different tasks. This paper has many contributions from both technical and conceptual perspectives. I applaud the authors' efforts to clearly categorized every algorithm according to its own features.

**Weaknesses:**

As the authors imply, they still lack some other SSL tasks such as imbalanced semi-supervised learning, open-set semi-supervised learning, etc. Although the authors use 14 different algorithms to conduct experiments, there are still some algorithms remaining. The table used to store experiment data seems too small, like tables 5, 6, and 7.



**Additional Feedback:**

None.

**Correctness:**

Yes, I think the evaluation methods and experiment design appropriate and performed correctly.

**Documentation:**

There is sufficient detail for reproduction.

**Ethics:**

None.

**Relation To Prior Work:**

This work clearly discusses the differences between previous work.

**Summary And Contributions:**

The authors notice that popular SSL evaluation protocols are often constrained to computer vision tasks and are generally time-consuming and environmentally unfriendly. Based on this, they construct USB by selecting several tasks on which they evaluate dominant semi-supervised learning(SSL) methods. To reduce the cost of SSL, they also provide a pre-training and fine-tuning paradigm. The used tasks are well divided according to their domains and algorithms are categorized.

---

> ### Author Response · Authors · 2022-08-17
> **Response to Reviewer ffrp**
>
> Thank you for the thoughtful comments. We now answer your questions.
>
> **Weakness 1:** As the authors imply, they still lack some other SSL tasks such as imbalanced semi-supervised learning, open-set semi-supervised learning, etc. Although the authors use 14 different algorithms to conduct experiments, there are still some algorithms remaining.
>
> **Answer:**
> We are well aware of the great challenges brought by limited labeled data such as in imbalance and open-set settings. In the current version of USB, the focus is on *traditional* semi-supervised setting by building a comprehensive and common baselines for researchers. Other tasks might be more challenging and need more tailored algorithms. USB is not a static project, but will iterate over time with our community after released. We plan to include increasingly more benchmarks in the future versions by extending with more algorithms and tasks.
>
> **Weakness 2:** The table used to store experiment data seems too small, like tables 5, 6, and 7.
>
> **Answer:**
>  We have re-organized the table in this revision. Please see the latest version of the paper.

---

### Author Response · Authors · 2022-08-26
**Summary of Rebuttal-Part1**

We thank all reviewers for their thoughtful and insightful reviews. We have read through them thoroughly and responded accordingly. **We fix some typos in our responses today and split the appendix and put it in the supplementary material.** Note that changes in the revised paper are highlighted in blue. Here we would like to highlight the primary responses/modifications we made according to reviewers' comments.

1. **Motivation**  We improved our writing to make our motivation clear. There are mainly two reasons why we focus beyond the CV area: **First**, from the application perspective, researchers in both NLP and Audio communities are getting more and more interested in SSL [1,2,3,4]. However, only a few algorithms such as UDA have been discussed in NLP, and it remains uncertain whether other algorithms such as FlexMatch and SimMatch work also. In a nutshell, there still lacks discussions about whether SSL methods that work well on CV tasks still work on NLP and Audio tasks. To answer the above question, we aim to build a unified SSL benchmark to provide intuitive cognition on all SSL algorithms across different domains. **Second**, from the SSL algorithm research perspective, different domains of datasets certainly pose different challenges to the algorithms, thus, designing a diverse testbed will help us develop more robust algorithms and help researchers better analyze their approaches to different datasets. While CV datasets are widely used in most ML algorithms, we think adding NLP and Audio datasets will enlarge that diversity and become a better testbed.
2. **Related works.** We have included a related work section in the revised paper.
3. **Re-scope the paper.** We toned down the paper by modifying the title to "USB: A Unified Semi-supervised Learning Benchmark for Classification" and made it clear in the revised paper that "unified" means the unification of domains. Besides, we provided semi-supervised regression results in the Appendix and discussed the Limitation of our work in the main content. Designing a unified benchmark for all kinds of tasks across all domains is very challenging and certainly beyond the scope of this one single paper. But we are committed to continuing working under USB to extend it to imbalanced SSL, openset SSL, Seq2Seq SSL, etc.
4. **Pre-trained backbone selection.** We provided additional experiments using different backbones for CV, NLP, and Audio tasks both in the response and revised paper. Besides, as a long-term project, we will update the benchmark regularly to new backbones regularly.
5. **Insights or explanations for the experimental results.** We have expanded the discussion in our revision. To our humble knowledge, we are the first to discuss whether current SSL methods that work well on CV tasks still work on NLP and Audio tasks. The insights behind different SSL methods across different domains should be an important future direction.
6. **Pretrained backbone selection.** We provided additional experiments using different backbones for CV, NLP, and Audio tasks both in the response and revised paper. Besides, as a long-term project, we will update the benchmark regularly to new backbones regularly.
7. **The choice of metrics.**  We have included mean error rate as a metric in addition to rank.

---

> ### Author Response · Authors · 2022-08-26
> **Summary of Rebuttal-Part2**
>
> 8. **ImageNet results.** We excluded the ImageNet results from the first version of the USB paper because it is time-consuming to train even with acceleration settings in USB. In the revision, we have included the ImageNet results in Appendix and encouraged the community to report ImageNet results since it is a reasonable dataset for hill-climbing.
> 9. **GAN results.** We have included additional results of the GAN-based algorithm in the responses to answer the questions about why we excluded the GAN-based algorithm in USB.
> 10. **Correlation between TorchSSL and USB.** We analyzed the correlation between TorchSSL and USB, and demonstrated that using different backbones and pre-training indeed affects the final rank of SSL algorithms.
> 11. **Maintenance plan.**  We made it clear in the revision that Microsoft Research Asia (MSRA) will provide both the support and resources for evaluations of new algorithms. Additionally, our team has been working closely via remote services during the development of USB, and **we certainly will continue expanding the collaboration to the whole SSL community just like other popular open-source communities**.
>
> **Finally, yet importantly, dear reviewers, thank you so much for devoting much time to making USB better. We are willing to discuss USB with you more deeply before the rebuttal ends :)**
>
> [1] Chen, Jiaao, Zichao Yang, and Diyi Yang. "MixText: Linguistically-Informed Interpolation of Hidden Space for Semi-Supervised Text Classification." Proceedings of the 58th Annual Meeting of the Association for Computational Linguistics. 2020.
>
> [2] Li, Changchun, Ximing Li, and Jihong Ouyang. "Semi-Supervised Text Classification with Balanced Deep Representation Distributions." Proceedings of the 59th Annual Meeting of the Association for Computational Linguistics and the 11th International Joint Conference on Natural Language Processing (Volume 1: Long Papers). 2021.
>
> [3] Lu, Kangkang, et al. "Semi-Supervised Audio Classification with Consistency-Based Regularization." INTERSPEECH. 2019.
>
> [4] He, Gewen, et al. "Image2audio: Facilitating semi-supervised audio emotion recognition with facial expression image." Proceedings of the IEEE/CVF Conference on Computer Vision and Pattern Recognition Workshops. 2020.

---

### Meta-Review · Area_Chair_togA · 2022-09-07

**Recommendation:** Accept
**Confidence:** 5

**Metareview:**

This article observed an active rebuttal period and definitely went to a final form that should be published at this venue, as highlighted by the latest updates of all reviewers. It will be a good add to the broad (multi domain) semi-supervised community!

---

### Decision · Program_Chairs · 2022-09-16

Accept